# Development of Ecosystem for Corporate Green Innovation: Resource Dependency Theory Perspective

**Daquan Gao** [1,2], **Christina W. Y. Wong** [1] **and Kee-hung Lai** [3,*]

1    Business Division, School of Fashion and Textiles, The Hong Kong Polytechnic University, Hong Kong, China
2    School of Management, Harbin Institute of Technology, Harbin 150001, China
3    Department of Logistics and Maritime Studies, The Hong Kong Polytechnic University, Hong Kong, China
*    Correspondence: mike.lai@polyu.edu.hk

**Abstract:** Although research on green innovation has increased dramatically in recent years, little is known about the system mechanisms for the innovation. Grounded in the resource dependency theory (RDT) and national innovation system (NIS) literature, this study examines the ecosystem in promoting green innovation from both theoretical and empirical perspectives. This study constructs a Nash-Cournot equilibrium to address the effects of national green innovation systems on corporate green innovation. Using data of 2136 A-share listed enterprises, 31 provincial-level R&D data, and 329 prefectural-level government annual work reports in China, this study finds that corporate collaboration, government subsidies, regional university R&D intensity, long-term credit, and government attention enhance the number of green innovation patents and patent diversity. Government attention strengthens the positive effects of corporate R&D cooperation, government subsidies, R&D intensity of regional research institutions, and long-term credit on the number of green innovation patents.

**Keywords:** government attention; green innovation; national innovation system; Nash-Cournot equilibrium; corporate collaboration; industry-academia collaboration

## 1. Introduction

Environmental issues have received substantial attention in corporate green operations [1,2]. To achieve green operations, an increasing number of companies are focusing on *green innovation* [3], which is referred to as innovation to address environmental issues and achieve sustainable development. Green innovation includes green product design and process innovation to enable energy savings, pollution prevention, and waste recycling [4]. Green innovation helps enterprises mitigate environmental risks [5], promote resource efficiency [6], reduce pollution rates [7,8], improve environmental performance [9], and improve ecological reputation [10]. Therefore, enterprises can benefit from green innovation by addressing global environmental issues to promote corporate development [11].

Beyond corporate behavior impact on green innovation [12,13], scholars have also studied the influence of external institutions on corporate green innovation, environmental regulation [14,15], government subsidies [16], and financial institutions [17,18]. Information exchange and coordination with external institutions are favorable for green innovation, allowing individual enterprises to engage in complex technological innovation with fewer capacity concerns and reduce technological and market uncertainties in green innovation activities [19,20]. Extant research focuses on the impact of internal organizational capabilities on green innovation or the impact of interactions between enterprises and external institutions on green innovation [21]. A single institution cannot perform all tasks of a green innovation ecosystem and requires multiple actors to collaborate for innovation [22]. Green innovation ecosystems are viable ways to collaborate to nurture a heterogeneous green value proposition for participants [23]. Previous studies on green innovation ecosystems include the game between enterprises and upstream and downstream enterprises [24,25],

the government-university-industry cooperative alliance [23], and the interaction mechanism between external environmental regulation and corporate internal green innovation processes [26]. Despite the existence of studies on green innovation systems, little is known about the government's influence on participation in green innovation.

This study addresses this research gap by constructing a green innovation ecosystem based on resource dependence theory (RDT) to examine the roles of government, the contribution of participants, and the influence of government in the innovation ecosystem. RDT emphasizes that enterprises obtain resources for sustainable development through interdependence and interaction with their surroundings [27]. Innovation in collaboration with other enterprises or institutions to form an innovation ecosystem is essential for resource access. A national innovation system (NIS) is an innovation network formed by the collaborative innovation of various sectors and institutions within a country [28]. Extant research mainly considers NIS as an innovation system that includes government, academia, and enterprises [29] and lacks attention to the complementarity between NIS and financial institutions [30]. Considering the large upfront investment in green innovation [31], high output uncertainty [32], and double externality [33], financial support from an external party is crucial for firms to engage in green innovation. Therefore, this study introduces financial institutions to research the national green innovation systems. Specifically, the innovation ecosystem is viewed as an entity comprising the government, research institutions (universities), enterprises, and financial institutions to increase corporate green innovation for national ecological development.

In this innovation ecosystem, the government provides subsidies for enterprises' green innovation, banks provide financial support for green innovation, and enterprises cooperate with research institutions in industry-academia-research to provide technical support and resource allocation for green innovation. Furthermore, the government provides guidance to enterprises, research institutions, and financial institutions through resource allocation [34] and signaling effects [35]. Considering that government policy takes a long time to draft, develop, and implement, as well as the regional differences for enforcement, the role of government is hardly recognized in a timely and adequate manner in the national green innovation system [36]. There is a concern regarding the process by which managers selectively focus on certain information while ignoring other parts [37]. Attention is an important part of governmental decision-making, arguing that government resource allocation and priorities change as the attention of policymakers shifts [38]. The annual work report of the local government summarizes the government's work in the past year and discloses its priorities in the coming years. Therefore, government attention measured by government work reports is more flexible than government policies, which can also reflect regional differences [39]. The frequency of words related to ecological development, green innovation, and innovative talent in government work reports reflects the extent to which the government pays attention to green innovation.

The main contributions of this study are threefold: First, it analyzes the roles of enterprises, governments, research institutions, and financial institutions in the national green innovation system, which provides a reference for countries keen on ecological problems and green development. Second, this study examines collaborative innovation among governments, financial institutions, research institutions, and enterprises, and how to increase corporate green innovation from an innovation system perspective. The government's role in developing a green innovation system to influence other participants in green systems is also highlighted. This study offers theoretical insights into the adoption of governmental actions to realize the collaborative innovation of multiple participants in green innovation ecosystems. Finally, this study develops a series of game models to analyze the national green innovation system. The game models examine the effects of financial markets, corporate collaboration innovation, industry-academia-research cooperation, government subsidies, and government environmental regulation on corporate green innovation. This study substantiates the role of national green innovation systems in promoting green innovation through analysis using game models and empirical methods.

## 2. Theory and Hypotheses Development

### 2.1. Resource Dependency Theory and Collaborative Innovation

RDT considers a firm an open system that depends on contingencies in the business environment [40], recognizing the influence of contextual constraints and conditions. This theoretical perspective is helpful for understanding collaborative innovation in enterprises [27,41]. Logsdon [42] argued that firms are motivated by high risk and interdependence for collaborative innovation based on RDT. Rennings [43] points out that green innovation has external environmental costs in addition to the spillover effects of innovation. The double externality of green innovation implies increased uncertainty in spontaneous corporate green innovation activities [44]. Therefore, external resources based on RDT play an important role in promoting green corporate innovation. For enterprises, universities have rich knowledge accumulation, many high-quality talents, advanced experimental equipment, and scientific research means, and strong scientific research ability. The government formulates innovation policies with the function of public management and services and supports innovation activities through government expenditures. Financial institutions have the advantages of securing capital resources and providing financial support for enterprises to pursue innovation activities.

### 2.2. National Green Innovation System

National innovation systems (NIS) are networks of public and private institutions located or rooted at the borders of nation-states whose activities and interactions initiate, import, modify, and disseminate new technologies [45]. According to the concepts of NIS and green innovation, a national green innovation system refers to the flow of resources, technology, and knowledge regarding green innovation between different institutions within a country to achieve green development. Analyzed from the historical development process, the advantages of developed countries such as the United States [46], the United Kingdom [47], and Japan [48] lie in the possession of an efficient and collaborative NIS. The NIS concept has been applied to developing countries, such as Brazil, India, China, and Thailand [49]. For developing countries, NIS is an important way to achieve the transformation of national industrialization [50]. The NIS literature highlights the importance of the interaction between different players in the economy to succeed in innovation.

According to the RDT, enterprises obtain resources from the government, financial institutions, research institutions, and other enterprises to increase green innovation. Figure 1 illustrates a national green innovation system. First, because the externality of innovation makes the competitive market structure less effective in motivating corporate innovation [51], collaborative innovation is suggested as a method of internalizing the external effects of innovation [52]. Enterprises share resources for innovation and collaborate to develop new technologies or products with the results shared by participants. Considering that enterprises need large capital investments to carry out green innovation, informal finance is an effective way for enterprises to carry out external financing [53]. Commercial credit is a flexible and convenient source of informal financing [54]. Second, collaboration between enterprises and universities or research institutions is a way for enterprises to gain access to unique resources such as technology, knowledge and capabilities, and the resulting collaborative innovation allows companies and universities to leverage each other's complementary skills, helpful to save costs and improve research results [55,56]. Third, for the common financing problem in the green innovation process of enterprises, banks provide loans to enterprises, which is beneficial to their green innovation [57]. Finally, the government mainly performs service functions, such as macro-regulation through environmental regulation [58–60], financial support for green innovation subjects through government subsidies [61,62]. The government has resources conducive to supporting innovation and regulating the resource allocation of each innovation participant [63]. Policies and regulations are generally developed by national ministries, while the prefectural-level government are responsible for their implementation [64]. Government attention at the local and municipal level lacks legal force in comparison with the policies and regulations

at the national level [65]. Specifically, environmental governance is primarily exercised by the Environmental Protection Agency (EPA), a subdivision of local government [66]. Local EPAs are weak, have limited resources, and therefore have weak legal effectiveness [67]. Government attention on green innovation, environmental protection, and talent can reflect the allocation of government resources, but given the lack of legal effect of prefectural-level governments, government attention has a moderating effect on the relationship of the innovation system participants on green innovation [68]. Government attention can strengthen the positive impact of the participants in the green innovation system (i.e., banks, research institutions, universities, etc.) on green innovation [69].

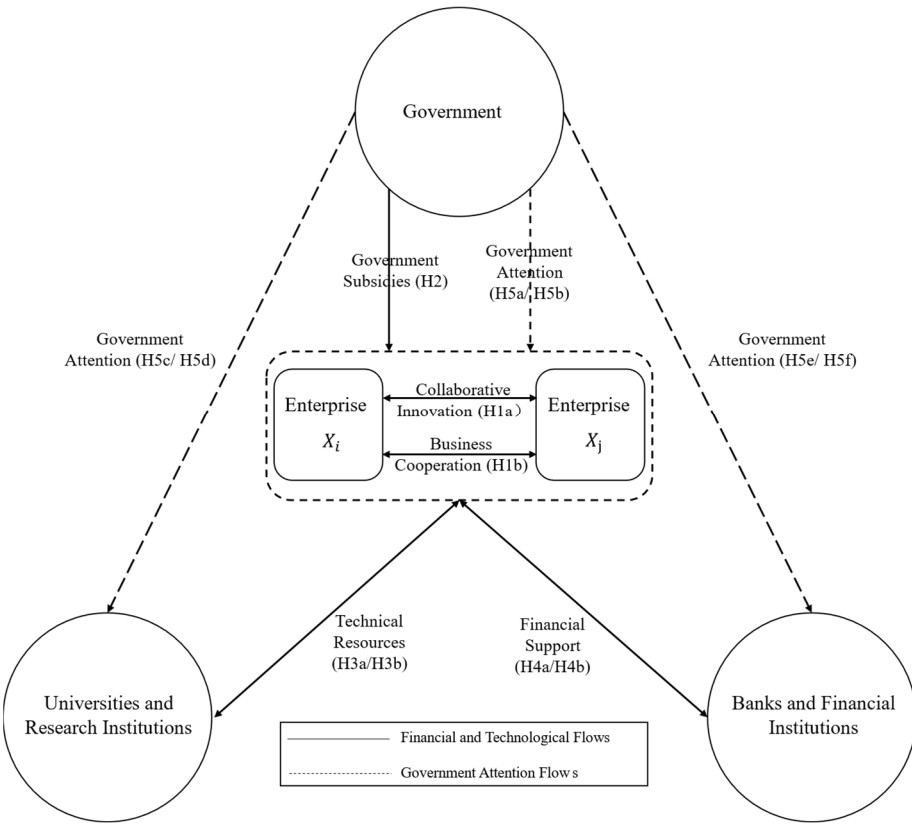

**Figure 1.** National Green Innovation System.

### 2.3. Corporate Collaboration and Corporate Green Innovation

In addition to relying on enterprises' resources to increase green innovation, collaboration with other firms is another important driver of green innovation [70,71]. Enterprises accumulate green innovation experience, improve their knowledge absorption capacity, and transform into green productivity in collaborative innovation. The key reasons why collaborative innovation increases green innovation in enterprises are elaborated below.

First, collaborative innovation helps enterprises acquire the technology, knowledge, information, and other resources needed to conduct green innovation activities and improve their core competitiveness [72,73]. Second, collaborative innovation with different types of enterprises is conducive to technological synergy and reduces the uncertainty in green R&D activities [74,75]. Third, collaborative innovation enables enterprises to absorb knowledge spillover from different enterprises, especially those with high technology, which can inject more intellectual capital into corporate green innovation [76,77]. Finally, as corporate green innovation has public interests, corporate cooperation can improve the exclusivity of corporate innovation results [78]. Many empirical studies have shown that collaboration is essential for green innovation [19,20]. Therefore, this study argues that enterprises that have granted patents that are jointly applied with other enterprises or institutions can

better carry out corporate green cooperative innovation, which is beneficial to corporate green innovation.

**Hypothesis 1a (H1a):** *Enterprises that have jointly granted patents with other enterprises or institutions are more likely to achieve corporate green innovation.*

Good relationships with companies include corporate and commercial credit. Commercial credit is a borrowing and lending relationship between companies formed by deferred payments or revenue received in advance in a commodity transaction [79,80]. Specific forms of commercial credit include accounts payable, notes payable, and advances from customers. Commercial credit alleviates financial pressure on enterprises [53,81]. Green innovation requires a large capital investment. Although commercial credit does not directly finance green innovation, it alleviates enterprises' cash pressure to invest capital in green innovation. Yu et al. [18] find that financing constraints measured by inverse indicators, such as accounts payable, are negatively associated with green innovation; that is, accounts payable are positively associated with corporate green innovation.

**Hypothesis 1b (H1b):** *Corporate commercial credit is positively associated with corporate green innovation.*

### 2.4. Government Subsidies and Corporate Green Innovation

The impact of government subsidies on firms' green innovation occurs mainly through three channels: financial support, signaling, and resource acquisition. First, considering the double externality of green innovation, it is difficult for firms to reach an optimal green innovation level [82,83]. Government innovation support may provide financial support for firms' green innovation. These subsidies increase the capital resources of enterprises for green innovation. In addition, government subsidies compensate for indirect losses caused by knowledge spillovers during R&D activities [84]. Second, government innovation support can send positive signals to firms that plan their development. Such subsidies also play a leading role in green innovation, and these positive signals can promote corporate improvement [85,86]. Finally, government innovation support improves firm access to resources. Government subsidies positively consolidate resources and reduce risks [87,88]. This kind of resource allocation can reduce the risk of green innovation for companies in the process of green innovation [89].

**Hypothesis 2 (H2):** *Government subsidies are positively associated with corporate green innovation.*

### 2.5. Research Institutions and Corporate Green Innovation

Universities and research institutions are essential for supporting a country's research and innovation development and are the cornerstones of national science and technology innovation. Science and technology innovation at universities and research institutions can be purposeful for green economic development. Universities and research institutions conduct research in green economic development to promote the transformation of the national economic development model, especially in green, environmental, energy-saving, new energy, ecological, and resource recycling industries. According to RDT, university-industry collaboration is an innovation driver. Knowledge and technology transfer between companies and university research institutions is expected to stimulate innovation because such collaboration combines not only heterogeneous partners, but more importantly, heterogeneous knowledge [90].

Some scholars have empirically confirmed the positive role of universities and research institutions in corporate green innovation, arguing that a green innovation ecosystem comprising research institutions and firms can stimulate green technological innovation based on its uncertainty characteristics in terms of cost investment, R&D risks, and economic effects [91,92]. Alliance-based industry-research institution research plays a significant role

in the green innovation ecosystem, and industry-research institution research collaboration has become an important alliance for achieving clean technology innovation development [23]. Prior studies have examined the impact of industry-research collaboration, but few have considered the spatial nature of innovation. Innovation is characterized by a clear spatial correlation [93]. Economic geography scholars argue that regional innovation output is inevitably influenced by local innovation inputs [94]. Lu et al. [95] found significant spatial clustering of green innovations, demonstrating that interregional innovation interactions increase regional green innovation outputs. The number of research institutions or universities, researchers, research outputs, and so on can be used as a measure of regional institutions and universities' research intensity. This study will examine the influence of universities and research institutions on corporate green innovation considering the regional research intensity of universities and research institutions.

**Hypothesis 3a (H3a):** *Regional university research intensity is positively associated with corporate green innovation.*

**Hypothesis 3b (H3b):** *The research intensity of regional research institutions is positively associated with corporate green innovation.*

### 2.6. Financial Institutions and Corporate Green Innovation

Enterprise green innovation often requires significant capital investment and a long developmental cycle [96]. An enterprise's R&D investment alone often cannot meet demand, which requires financial institutions to provide liquidity for enterprises and support their innovation activities [18]. Therefore, based on RDT, bank credit, as an important form of external financing, is favorable for corporate green innovation [97].

Previous studies have confirmed the positive impact of bank loans on corporate green innovation from both theoretical and empirical perspectives. Huang et al. [16] analyzed the promotion effect of loans on corporate green innovation using a game theory model. Yu et al. [18] empirically confirm the positive impact of corporate loans on green innovation. Ghisetti et al. [98] found that perceptions of financial barriers hinder firms' environmental innovation and suggest stimulating the adoption of ecological innovation by facilitating firms' access to bank loans. However, only a few scholars have studied the impact of loan maturity on green corporate innovation. Owing to the high repayment pressure of short-term loans, enterprises face difficulties in capital turnover, weakening their willingness to engage in green innovation. By contrast, when long-term loans are available, the repayment pressure is reduced, thus facilitating enterprises to conduct green innovation and increase their green innovation capacity.

**Hypothesis 4a (H4a):** *Short-term loans are negatively associated with corporate green innovation.*

**Hypothesis 4b (H4b):** *Long-term loans are positively associated with corporate green innovation.*

### 2.7. The Moderating Role of Government Attention

To better understand the role of government in national green innovation systems, this study introduces the concept of attention. This concept originated from the psychology field, which was later introduced into management studies and is defined as the process by which managers selectively focus on specific information while ignoring other parts [37]. The application of government attention to the study of national green innovation systems is highly innovative. It represents the attention of government decision-makers to specific matters; therefore, attention is also scarce. Government attention varies across political systems. In the United Kingdom, the Queen's speech is crucial for providing a general statement about executive priorities [99]. The Queen's speeches help governments build legislative plans, define policy decisions, and express and justify policy preferences [100]. The Government Annual Report in China is analogous to the Queen's speech [101].

According to RDT, government attention is a scarce resource for corporate green innovation. As governments become more environmentally concerned, policymakers allocate their limited attention to solving environmental problems [102]. Accordingly, government resource allocation changes according to the agenda [103]. The government's annual report is a guiding document that typically outlines the government's achievements over the past year and sets strategic goals for the following year based on specific factors, such as GDP development, environmental protection, and social security [104]. The number of textual phrases and critical terms in annual government reports reflects decision makers' attention to public affairs. Policies and regulations are generally formulated by national ministries and commissions, and each municipal government is responsible for their implementation [64]. Therefore, relative to national-level policies and regulations, government attention at the prefecture level lacks legal effect. Government attention plays a guiding role for enterprises, banks, universities, and research institutions in the national green innovation system, promoting knowledge and technology transfer from research institutions and universities to enterprises, improving the quality of green innovation services of banks and financial institutions, and driving green collaborative innovation among all subjects in the system.

**Hypothesis 5a (H5a):** *Government attention strengthens the effect of corporate collaboration on corporate green innovation.*

**Hypothesis 5b (H5b):** *Government attention strengthens the impact of commercial corporate credit on green innovation.*

**Hypothesis 5c (H5c):** *Government attention strengthens the effect of government subsidies on corporate green innovation.*

**Hypothesis 5d (H5d):** *Government attention strengthens the effect of regional universities' R&D intensity on corporate green innovation.*

**Hypothesis 5e (H5e):** *Government attention strengthens the impact of research institutions' R&D intensity on corporate green innovation.*

**Hypothesis 5f (H5f):** *Government attention strengthens the impact of bank loans on green corporate innovation.*

### 3. Data and Research Methodology

*3.1. A Market Equilibrium Model of National Green Innovation System Based on Nash-Cournot Equilibrium Theory*

Game theory is viewed as a mathematical model of strategic interactions between independent agents that provides decision makers with prescriptions and recommendations to help develop and implement effective strategies [105]. Therefore, this study constructs a Nash-Cournot game model that involves enterprises, governments, financial institutions, and research institutions.

Consider an industry with $N$ firms facing an inverse demand function

$$P = a - bQ;\ Q = \sum_{i=1}^{N} q_i \text{ with } a, b > 0$$

The cost function of the firm is

$$C(Q) = cq_i;$$

$c$ is the fixed cost per unit.

The profit function for firms 1 and 2 for Period 1 is

$$\pi_i = [a - bQ]q_i - cq_i$$

According to the Nash-Cournot equilibrium $\frac{\partial \pi_i}{\partial q_i} = 0$.

$$q_i^* = \frac{a - c}{(N+1)b}; \ Q^* = \frac{(a - c)N}{(N+1)b}$$

$$P^* = \frac{a + Nc}{N+1}; \ \pi_i^* = \frac{(a - c)^2}{b(N+1)^2}$$

In Period 2, firm $i$ begins to engage in green innovation, and the effective R&D investment of firm $i$ consists of the firm's R&D investment and R&D spillover from other firms [106].

$$R_i = r_i + \beta \sum_{i \neq j} r_j$$

$R_i$ is the effective R&D investment for firm $i$; $r_i$ is the individual R&D investment for firm $i$; $\beta$ is an R&D spillover parameter. $0 \leq \beta \leq 1$.

The green innovation output function is positively correlated with R&D input, but considers the marginal decreasing effect of R&D output. The green innovation output function is monotonically increasing and concave [16]; thus, the innovation output function is:

$$k_i = \left(1 - e^{-R_i}\right)$$

The cost of company $i$ at this point is:

$$c_i' = c + e^{-R_i} * r_i$$

According to the Nash-Cournot equilibrium $\frac{\partial \pi_i}{\partial q_i} = 0$.

$$q_i'^* = q_i^* - \frac{e^{-R_i} * r_i}{(N+1)b}; Q'^* = Q^* - \frac{\sum_{i=1}^N e^{-R_i} * r_i}{(N+1)b}$$
$$P'^* = P^* + \frac{\sum_{i=1}^N e^{-R_i} * r_i}{(N+1)};$$

At this point, the price of the product in equilibrium is greater than the price in period 1 and the firm's output is less than the price in period 1. Since it is difficult for consumers to accept the price increase, the firm will still sell at the period 1 price. The firm's profit in period 2 is

$$\pi_i'^* = \pi_i^* - \frac{(a - c)e^{-R_i} * r_i}{(N+1)^2 b}$$

Considering the R&D spillover after a firm's green innovation, the firm's output and profits will be significantly lower in period 2.

Under the national green innovation system, enterprises are encouraged to conduct green cooperative R&D when their effective R&D investment is.

$$R_i = \sum r_i$$

Consider the equilibrium conditions for firm $i$ under a national green innovation system.

$$c_{NGIS}' = c + t * e^{-\theta(R_i + w_i)} + e^{-\theta(R_i + w_i)} * r_i - w_i - \left(1 - e^{-\theta(R_i + w_i)}\right) * w_i$$

$$w_i = s_i + l_i$$

$t$ is the environmental tax charge, and $t$ is negatively related to R&D investment; $w_i$ is the external input to corporate green innovation for firm $i$, including direct government subsidies $s_i$, green credit from financial institutions $l_i$. Considering that green credits are generally policy-oriented and have low interest rates, they are neglected. $\theta$ is the increment of industry-research cooperation on green enterprise innovation, $\theta > 1$.

At this point, the equilibrium price under the national green innovation system is

$$P^*_{NGIS} = P^* + \frac{\sum\left[(t + r_i + w_i)e^{-\theta(R_i + w_i)} - 2w_i\right]}{N + 1}$$

While government regulations increase firm costs in a national green innovation system, government subsidies, green credit, and internal collaboration reduce firm costs. When the external inputs of firm are higher than the internal inputs of firm, that is, $w_i$ is greater than $(t_i + r_i)$, the equilibrium market price is lower. The higher the enhancement brought about by industry-research cooperation for corporate green innovation, the lower the equilibrium market price. Therefore, the market will be put into production at a new equilibrium price and consumers will buy products with lower inputs. Both consumers and manufacturers benefit from a national green innovation system.

### 3.2. Sample Selection and Data Sources

China is a major energy-consuming country; however, in recent years, it has committed to ecological development. Although the concept of ecological civilization construction was first proposed at the 17th Congress of the Communist Party of China (CPC) in 2007, the positioning of the circular economy was further enhanced at the 18th CPC Congress in 2012. The report of the 19th Party Congress further calls for the establishment of a sound economic system of green, low-carbon, and circular development to promote the overall green transformation of China's economy and society. China has elevated its ecological initiatives to the level of a national strategy, and green innovation is no longer just a corporate-level issue. Simultaneously, China has made great strides in talent development, scientific research, and socio-economic development. Therefore, it is informative to take China as an example to establish a national green innovation system.

Using China as a research sample can provide lessons for other countries pursuing ecological governance. After excluding special treatment companies that have financial abnormalities with the risk of delisting, companies listed later than 2011, companies in the financial sector, and 2136 Chinese A-share listed companies in the Shanghai and Shenzhen exchanges from 2011 to 2020 were selected for this study.

All enterprise-level data sources used in this study are obtained from the WIND data terminal. WIND is a comprehensive database focusing on Chinese finance and economics. Enterprise Green Patent Data were obtained from the WinGo platform. WinGo Green Patent Database provides listed enterprises with green patent information, mainly the number of green patent applications and grants according to the World Intellectual Property Organization (WIPO) patent classification and Organization for Economic Cooperation and Development (OECD) patent classification. Government attention data was obtained from government work reports and the National Bureau of Statistics. WinGo platform was used for text analysis. We analyze the text data processing in three steps: (1) Download the government work report and government statistical yearbook from the government website. (2) Use WinGo text data platform to conduct text analysis of the government work report and calculate the frequency of words related to government environmental protection, green innovation, and talent concern. (3) Use Stata to process the data and perform empirical regression analysis. R&D data from regional universities and institutions was obtained from the China Science and Technology Statistical Yearbook.

### 3.3. Variables

3.3.1. Measure of Government Attention

Text analysis is a popular method for measuring government attention and has been widely used in many areas of social science research as a combination of qualitative and quantitative research methods [107]. This study analyzes the text of the work reports of 329 Chinese prefectural governments from 2011 to 2020 and identifies ecological environment attention, green development attention, basic research, and scientific and technological talent attention. Government attention was measured by the number of words and sentences related to ecological development, green innovation, and innovation talent concerns in prefecture-level government annual work reports. The specific process of text analysis inputs the initial word set, analyzes similar words, expands the word set, and obtains the word frequency.

3.3.2. Measure of Regional Universities and Research Institution R&D Capacity

This study selected the number of research institutions in a province, researchers, internal R&D expenditures, R&D projects, and research output, such as papers and patents, to construct the regional research intensity index of the research institution. This study selected the number of universities in the province, university R&D personnel, internal R&D expenditure, R&D projects, and research output to construct regional university research intensity.

3.3.3. Measure of Green Innovation

Patents are a typical proxy for innovation output, and many scholars use the number of patents granted as a direct indicator of a firm's innovation output [13,108]. In addition, some scholars have used the Herfindahl-Hirschman Patent index (HHI) to measure the concentration of a firm's innovation activities and argue that firms need to enhance their core competencies and focus on a specific technology area to deepen their technological activities and improve their specialization [109,110]. However, for green innovation, a wide range of innovations involving multiple fields can achieve the strategic goal of corporate green development. Therefore, this study considers that patent dispersion can better reflect the quality of corporate green innovation.

This study examines green innovation from two perspectives: green patent granted data, including the number of green patents granted and the number of green invention patents granted, and enterprise green patent diversity data. In this study, the Herfindahl-Hirschman index (HHI) method at the group level was adopted to calculate green patent diversity.

$$Patent\_Diversity = \begin{cases} (1 - \sum \alpha^2) + 1, \ if \ grant \ patent \neq 0 \\ 0, \ if \ grant \ patent = 0 \end{cases}$$

The World Intellectual Property Organization (WIPO) launched a tool designed to facilitate the retrieval of patent information related to environmentally friendly technologies, dividing green patents into seven subcategories: alternative energy production, transportation, energy conservation, waste management, agriculture and forestry, administrative regulation and design, and nuclear energy. $\alpha$ is the percentage of green patents for each company category.

3.3.4. Control Variables

To improve the accuracy of our results, we must minimize the internal and external changes that might impact company performance. We controlled for corporate management data, R&D investment, and profitability in the analyses. Board diversity is a key governance mechanism to improve corporate green performance. Following Jiang and Yuan [111], we include a series of corporate management data that impacts a firm's green innovation. Corporate management data include corporate equity concentration (percentage

of shares held by the top ten shareholders), percentage of institutional ownership, management salaries, agency costs, and board size. Additionally, corporate R&D investment and profitability can affect corporate green innovation [112]. Corporate profitability includes the ratio of corporate EBIT to total assets, return on assets, and operating margins. The variables used in the study are shown in Table 1.

**Table 1.** The variables used in the study.

| Variable Code | Variable Explanation |
|---|---|
| **Explained Variable** | |
| ISUM | The number of green invention patents granted |
| TSUM | The number of green patents granted |
| IPK | Diversity of green invention patents |
| TPK | Diversity of green patents |
| **Explanatory Variables** | |
| SUB | Government Subsidy |
| Bcredit | Ratio of accounts payable and prepayments received to the operating revenue of the enterprise |
| CB | Corporate has granted patents which are jointly applied with other corporate or institution |
| INST | Regional Research Institution R&D Capabilities |
| HE | Regional University R&D Capabilities |
| STDI | Annual short-term bank loan increment for enterprises |
| LTDI | Annual long-term bank loan increment for enterprises |
| **Moderating Variables** | |
| Government Attention | The number of occurrences of words and sentences related to ecological development, green innovation, and innovation talent concerns |
| **Control Variables** | |
| TOP | Percentage of shares held by the top ten shareholders |
| inst | Percentage of institutional ownership |
| MS | Management salaries |
| MB | Board size |
| TAT | Total asset turnover ratio |
| LNTA | Corporate size |
| RDE | Corporate R&D expenses |
| EBITR | The ratio of EBIT to total assets |
| ROA | Return on assets |
| OPTOGR | Operating margin |

*3.4. Research Methodology*

To test the influence of each participant in the green innovation system (i.e., banks, research institutions, universities, etc.) on corporate green innovation, i.e., Hypotheses 1–4, this study uses a panel data fixed effects model with the introduction of industry- and time-fixed effects [113].

$$Y_{i,t} = \alpha_0 + \alpha_1 X_{i,t} + \alpha_2 Z_{i,t} + IndFE + YearFE + \varepsilon_2$$

where $Y_{i,t}$ is the explained variable for firm $i$ at time $t$. $X_{i,t}$ is the explanatory variable for firm $i$ at time $t$. $Z_{i,t}$ is a vector of the firm-level control variables.

To test Hypotheses 5a–5f, this study uses the causal step technique [114]. The moderation model was tested using a causal step technique analysis as a way to eliminate the interaction between the control variables and the main variables [115].

$$Y_{i,t} = \alpha_0 + \alpha_1 M_{i,t} + \alpha_2 Z_{i,t} + IndFE + YearFE + \varepsilon_1$$

$$Y_{i,t} = \alpha_0 + \alpha_1 X_{i,t} + \alpha_2 M_{i,t} + \alpha_3 X * M_{i,t} + \alpha_4 Z_{i,t} + IndFE + YearFE + \varepsilon_3$$

where $Y_{i,t}$ is the explained variable for firm $i$ at time $t$. $M_{i,t}$ is the moderating variable for firm $i$ at time $t$. $X_{i,t}$ is the explanatory variable for firm $i$ at time $t$. $X * M_{i,t}$ is the interaction between the explained and moderating variables. $Z_{i,t}$ is a vector of the firm-level control variables.

## 4. Results

### 4.1. Descriptive Statistics

Table 2a,b describe the variables used in this study for the final sample of 21,360 firm-year observations between 2011 and 2020. There were 919 state-owned enterprises (SOEs) and 1217 non-state-owned enterprises (non-SOEs).

**Table 2.** (**a**): Descriptive statistics on key variables (All Sample). (**b**): Descriptive statistics on key variables (Non-state-owned enterprises and State-owned enterprises).

| (a) | | | | | | |
|---|---|---|---|---|---|---|
| | (1) | (2) | (3) | (4) | (5) | (6) |
| Variables | N | Mean | Median | STD | Min | Max |
| TSUM | 21,360 | 1.960 | 0 | 9.580 | 0 | 186 |
| ISUM | 21,360 | 0.520 | 0 | 2.820 | 0 | 53 |
| TPK | 21,360 | 0.300 | 0 | 0.560 | 0 | 1.810 |
| IPK | 21,360 | 0.140 | 0 | 0.390 | 0 | 1.670 |
| GOVATT | 21,360 | 88.86 | 86 | 26.44 | 30 | 174 |
| Bcredit | 21,360 | 0.310 | 0.230 | 0.290 | 0.0100 | 1.890 |
| CB | 21,360 | 0.330 | 0 | 0.470 | 0 | 1 |
| SUB | 21,360 | 0.390 | 0.110 | 0.900 | 0 | 8.180 |
| INST | 21,360 | 0.500 | 0.150 | 1.410 | −0.620 | 5.130 |
| HE | 21,360 | 0.850 | 0.770 | 1.090 | −1.430 | 3.520 |
| STDI | 21,360 | 1.040 | 0 | 8.870 | −47.73 | 69.89 |
| LTDI | 21,360 | 1.560 | 0 | 11.37 | −36.57 | 125.7 |

| (b) | | | | | | | | | | | |
|---|---|---|---|---|---|---|---|---|---|---|---|
| | Non-State-Owned Enterprises (N = 9190) | | | | | State-Owned Enterprises (N = 12,170) | | | | | *t*-Test |
| Variables | (2) | (3) | (4) | (5) | (6) | (7) | (8) | (9) | (10) | (11) | (2)–(7) |
| | Mean | Median | STD | Min | Max | Mean | Median | STD | Min | Max | Diff |
| TSUM | 2.360 | 0 | 11.17 | 0 | 186 | 1.660 | 0 | 8.160 | 0 | 186 | −0.703 *** |
| ISUM | 0.640 | 0 | 3.340 | 0 | 53 | 0.420 | 0 | 2.360 | 0 | 53 | −0.226 *** |
| TPK | 0.330 | 0 | 0.590 | 0 | 1.810 | 0.270 | 0 | 0.530 | 0 | 1.810 | −0.057 *** |
| IPK | 0.170 | 0 | 0.420 | 0 | 1.670 | 0.130 | 0 | 0.360 | 0 | 1.670 | −0.045 *** |
| GOVATT | 90.84 | 88 | 27.72 | 31 | 174 | 87.36 | 86 | 25.32 | 30 | 174 | −3.475 *** |
| Bcredit | 0.350 | 0.250 | 0.320 | 0.010 | 1.890 | 0.290 | 0.220 | 0.260 | 0.010 | 1.890 | −0.061 *** |
| CB | 0.390 | 0 | 0.490 | 0 | 1 | 0.280 | 0 | 0.450 | 0 | 1 | −0.113 *** |
| SUB | 0.520 | 0.150 | 1.100 | 0 | 8.180 | 0.280 | 0.090 | 0.680 | 0 | 8.180 | −0.242 *** |
| INST | 0.700 | 0.150 | 1.690 | −0.620 | 5.130 | 0.350 | 0.150 | 1.120 | −0.620 | 5.130 | −0.348 *** |
| HE | 0.780 | 0.660 | 1.150 | −1.430 | 3.520 | 0.910 | 0.790 | 1.030 | −1.430 | 3.520 | 0.125 *** |
| STDI | 1.280 | 0 | 11.31 | −47.73 | 69.89 | 0.860 | 0 | 6.440 | −47.73 | 69.89 | −0.418 *** |
| LTDI | 2.360 | 0 | 14.21 | −36.57 | 125.7 | 0.950 | 0 | 8.580 | −36.57 | 125.7 | −1.412 *** |

Please refer to Table 1 for an explanation of the variable. *** represent statistical significance at the 1% levels.

State-owned enterprises (SOEs) play a crucial role in China's economic development. By further analyzing SOEs and non-SOEs, we obtain the following findings: the total green patent output, green invention patent output, green innovation diversity, and green invention innovation diversity of SOEs are significantly higher than those of non-SOEs. SOEs also receive significantly more government attention than non-SOEs. Moreover, due to government endorsement, SOEs receive higher government subsidies, bank credit, and commercial credit than non-SOEs do. Compared with SOEs, the research capacity of universities in regions where non-SOEs are located is significantly more robust. Table 3 presents the correlation coefficients for the pertinent variables in the final sample of 21,360 observations. Multicollinearity is a difficult problem because the correlation coefficient between explanatory variables is less than 0.50.

**Table 3.** Correlation coefficients.

|  | TSUM | ISUM | TPK | IPK | GOVATT | Bcredit | CB | SUB | INST | HE | STDI | LTDI |
|---|---|---|---|---|---|---|---|---|---|---|---|---|
| TSUM | 1 | | | | | | | | | | | |
| ISUM | 0.833 *** | 1 | | | | | | | | | | |
| TPK | 0.438 *** | 0.400 *** | 1 | | | | | | | | | |
| IPK | 0.477 *** | 0.541 *** | 0.748 *** | 1 | | | | | | | | |
| GOVATT | 0.064 *** | 0.066 *** | 0.020 *** | 0.043 *** | 1 | | | | | | | |
| Bcredit | 0.036 *** | 0.022 *** | 0.050 *** | 0.033 *** | 0.053 *** | 1 | | | | | | |
| CB | 0.183 *** | 0.183 *** | 0.322 *** | 0.287 *** | 0.055 *** | 0.017 ** | 1 | | | | | |
| SUB | 0.335 *** | 0.345 *** | 0.246 *** | 0.278 *** | 0.078 *** | 0.001 | 0.263 *** | 1 | | | | |
| INST | 0.091 *** | 0.088 *** | 0.040 *** | 0.067 *** | 0.509 *** | 0.048 *** | 0.072 *** | 0.138 *** | 1 | | | |
| HE | 0.066 *** | 0.063 *** | 0.048 *** | 0.063 *** | 0.372 *** | 0.028 *** | 0.057 *** | 0.046 *** | 0.639 *** | 1 | | |
| STDI | −0.013 * | −0.016 ** | 0.029 *** | 0.022 *** | 0.007 | 0.010 | 0.052 *** | 0.123 *** | 0.022 *** | 0.003 | 1 | |
| LTDI | 0.145 *** | 0.136 *** | 0.067 *** | 0.068 *** | 0.064 *** | 0.075 *** | 0.092 *** | 0.212 *** | 0.090 *** | 0.043 *** | 0.099 *** | 1 |

Please refer to Table 1 for an explanation of the variable. *, **, and *** represent statistical significance at the 10%, 5%, and 1% levels.

### 4.2. Empirical Results

Table 4a presents the panel OLS regression results for the impact of corporate collaboration, research institutions, universities, and financial institutions on corporate green innovation, and the moderating effects of government attention. All models include year and industry fixed effects.

**Table 4.** (**a**): Panel OLS regression of the national green innovation system. (**b**): The results of the hypothesis 1a to hypothesis 5f.

| | (a) | | | | | |
|---|---|---|---|---|---|---|
| | (1) | (2) | (3) | (4) | (5) | (6) |
| | TSUM | TSUM | TSUM | TPK | TPK | TPK |
| CB | 0.778 *** | | −0.522 | 0.198 *** | | 0.148 *** |
| | (5.909) | | (−1.236) | (25.328) | | (5.882) |
| Bcredit | 1.663 *** | | 3.706 *** | 0.155 *** | | 0.208 *** |
| | (7.104) | | (5.477) | (11.190) | | (5.166) |
| SUB | 0.750 *** | | −0.266 | 0.001 * | | 0.059 *** |
| | (9.179) | | (−1.199) | (1.663) | | (4.504) |
| INST | −0.024 | | −1.331 *** | −0.017 *** | | 0.016 |
| | (−0.455) | | (−5.280) | (−5.355) | | (1.093) |
| HE | 0.251 *** | | 0.568 ** | 0.027 *** | | −0.014 |
| | (3.743) | | (2.092) | (6.804) | | (−0.862) |
| STDI | −0.051 *** | | 0.093 *** | −0.000 | | −0.002 * |
| | (−8.055) | | (4.463) | (−0.523) | | (−1.747) |
| LTDI | 0.019 *** | | −0.067 *** | −0.002 *** | | −0.000 |
| | (3.787) | | (−4.152) | (−5.934) | | (−0.025) |
| GOVATT | | 0.014 *** | 0.007 * | | 0.000 *** | 0.001 ** |
| | | (6.204) | (1.788) | | (2.985) | (2.266) |
| GOVATT × CB | | | 0.015 *** | | | 0.001 ** |
| | | | (3.308) | | | (2.026) |
| GOVATT × Bcredit | | | −0.022 *** | | | −0.001 * |
| | | | (−3.143) | | | (−1.657) |
| GOVATT × SUB | | | 0.010 *** | | | −0.001 *** |
| | | | (4.922) | | | (−4.965) |
| GOVATT × INST | | | 0.009 *** | | | −0.000 ** |
| | | | (4.302) | | | (−2.522) |
| GOVATT × HE | | | −0.002 | | | 0.000 ** |
| | | | (−0.644) | | | (2.399) |
| GOVATT × STDI | | | −0.000 *** | | | 0.000 |
| | | | (−7.114) | | | (1.601) |
| GOVATT × LTDI | | | 0.000 *** | | | −0.000 * |
| | | | (5.551) | | | (−1.855) |
| _cons | 2.918 ** | −2.631 ** | 2.045 | −1.086 *** | −1.503 *** | −1.119 *** |
| | (2.134) | (−2.051) | (1.448) | (−13.395) | (−19.481) | (−13.335) |

**Table 4.** *Cont.*

| (a) | | | | | | |
|---|---|---|---|---|---|---|
| | **(1)** | **(2)** | **(3)** | **(4)** | **(5)** | **(6)** |
| | **TSUM** | **TSUM** | **TSUM** | **TPK** | **TPK** | **TPK** |
| control | yes | yes | yes | yes | yes | yes |
| industry | yes | yes | yes | yes | yes | yes |
| year | yes | yes | yes | yes | yes | yes |
| N | 21,360 | 21,360 | 21,360 | 21,360 | 21,360 | 21,360 |
| r2 | 0.312 | 0.305 | 0.319 | 0.295 | 0.266 | 0.297 |

| (b) | |
|---|---|
| **Hypothesis** | **Result** |
| 1a: Enterprises with granted patents jointly with other enterprises or institution are likely to achieve corporate green innovation. | Support |
| 1b: Corporate commercial credit is positively associated with corporate green innovation. | Support |
| 2: Government subsidies are positively associated with corporate green innovation. | Support |
| 3a: Regional university research intensity is positively associated with corporate green innovation. | No |
| 3b: Regional research institutions research intensity is positively associated with corporate green innovation. | Support |
| 4a: Short-term loans are negatively associated with corporate green innovation. | Support |
| 4b: Long-term loans are positively associated with corporate green innovation. | Support |
| 5a: Government attention strengthens the impact of corporate collaboration innovation on corporate green innovation. | Support |
| 5b: Government attention strengthens the impact of corporate commercial credit on corporate green innovation. | No |
| 5c: Government attention strengthens the impact of government subsidies on corporate green innovation. | Partially |
| 5d: Government attention strengthens the impact of R&D intensity of regional universities on corporate green innovation. | No |
| 5e: Government attention strengthens the impact of R&D intensity of research institutions on corporate green innovation. | Partially |
| 5f: Government attention strengthens the impact of bank loans on corporate green innovation. | Partially |

The *t*-statistics are in parentheses. * $p < 0.1$, ** $p < 0.05$, *** $p < 0.01$.

According to the results in Table 4a, a corporation with granted patents that are jointly applied with other corporations or institutions (CB) is positively associated with corporate green innovation output ($\beta = 0.778$, $p < 0.01$) and corporate green innovation patent diversity ($\beta = 0.198$, $p < 0.01$), supporting that corporate innovation collaboration is beneficial for corporate green innovation performance in the Nash-Cournot equilibrium model proposed in this study, and confirms the positive correlation between corporate innovation collaboration and corporate green innovation in the literature [73,77]. Good commercial credit also increased corporate green innovation output ($\beta = 1.663$, $p < 0.01$) and patent diversity ($\beta = 0.155$, $p < 0.01$). Government subsidies are positively related to firms' green innovation output ($\beta = 0.750$, $p < 0.01$) and green innovation patent diversity ($\beta = 0.001$, $p < 0.1$), which is consistent with the findings of previous studies [89]. Regional university R&D intensity is positively associated with corporate green innovation output ($\beta = 0.251$, $p < 0.01$) and green innovation patent diversity ($\beta = 0.027$, $p < 0.01$). However, regional research institutions' R&D intensity is not related to green innovation output or is negatively related to green innovation patent diversity ($\beta = -0.017$, $p < 0.01$). Currently, Chinese research institutions primarily undertake the government's science and technology innovation. As a result, the link between Chinese research institutions and enterprises is not strong, and investment in research institutions can crowd out the patent diversity between universities and enterprises. An increase in short-term corporate bank loans is not conducive to corporate green innovation output ($\beta = -0.051$, $p < 0.01$). The high repayment pressure of short-term loans and the uncertainty of corporate green innovation render short-term loans unfavorable to corporate green innovation. Long-

term corporate loans increase corporate green innovation output ($\beta = 0.019$, $p < 0.01$), but decrease corporate green innovation patent diversity ($\beta = -0.002$, $p < 0.01$). The repayment pressure for long-term loans by the enterprises was low. However, it still puts financial pressure on companies, which reduces patent diversity to lower the uncertainty of green innovation. In summary, hypotheses 1a, 1b, 2, 3a, and 4b were confirmed.

This study found that government attention directly increases corporate green innovation output ($\beta = 0.014$, $p < 0.01$) and corporate green innovation patent diversity ($\beta = 0.000$, $p < 0.01$). Through an analysis of the moderating effect of government attention, this study finds that government attention increases the positive impact of corporate innovation cooperation based on corporate green innovation ($\beta = 0.015$, $p < 0.01$; $\beta = 0.001$, $p < 0.05$); that is, it has a positive moderating effect. However, government attention weakens the positive impact of corporate commercial credit on corporate green innovation ($\beta = -0.022$, $p < 0.01$; $\beta = -0.001$, $p < 0.1$), that is, it has a negative moderating effect. Government attention strengthens the positive impact of government subsidies on corporate green innovation output ($\beta = 0.010$, $p < 0.01$) but also increases the negative effect of government subsidies on corporate green patent diversity ($\beta = -0.001$, $p < 0.01$). Although the R&D intensity of regional research institutions is not related to green innovation output, government attention alleviates this irrelevance ($\beta = 0.009$, $p < 0.01$). However, government attention strengthens the negative correlation between regional R&D intensity and green innovation patent diversity ($\beta = -0.000$, $p < 0.05$). Government attention strengthens the positive correlation between regional universities' R&D intensity and corporate green innovation patent diversity ($\beta = 0.000$, $p < 0.05$). Specifically, government attention strengthens the negative impact of short-term lending on the diversity of firms' green output ($\beta = -0.000$, $p < 0.01$). Government attention strengthens the positive impact of long-term borrowing on firms' green innovation output ($\beta = 0.000$, $p < 0.01$), but also strengthens the negative impact of long-term lending on firms' green innovation patent diversity ($\beta = -0.000$, $p < 0.1$).

The results of the hypotheses are summarized in Table 4b. Specifically, corporate collaboration, government subsidies, regional research institutions' research intensity, and long-term loans are positively associated with green corporate innovation. Short-term loans are negatively associated with corporate green innovation; that is, Hypotheses 1a, 1b, 2, 3a, 3b, 4a, and 4b. Government attention strengthens the impact of corporate collaborative innovation, government subsidies, research institutions, and bank loans on the number of green corporate patents. Government attention strengthens the impact of collaborative innovation on corporate green patent diversity. Government attention weakens the impact of corporate commercial credit, government subsidies, research intensity of regional research institutions, and long-term loans on corporate green patent diversity. Therefore, Hypothesis 5a is supported: Hypotheses 5c, 5e, and 5f are partially supported, while Hypotheses 5b and 5d are not supported.

*4.3. Heterogeneity Analysis*

Through continuous reform practices and theoretical exploration, Party organization has been added to the main structure of the corporate governance of modern Chinese enterprises. It is a party organization that strengthens the connection between state-owned enterprises and the government. Therefore, it is necessary to study the characteristics of SOEs in the national green innovation system separately. As part of our investigation and validation process, we conducted further analysis. Table 5 shows the heterogeneity analysis of SOEs and non-SOEs. For SOEs and non-SOEs, corporate collaboration, government subsidies, regional university R&D intensity, and bank credit have the same direction of influence on corporate green innovation output but to different degrees. Government attention positively affects the green innovation output of SOEs but has no significant impact on non-SOEs. The moderating effect of government attention affects SOEs and non-SOEs differently. Government attention can strengthen the negative impact of short-term loans on corporate green innovation output for both SOEs and non-SOEs. Government

attention moderates the positive impact of regional R&D intensity and long-term loans on corporate green innovation in SOEs. Government attention strengthens the positive impact of government subsidies on green innovation output for SOEs but not for non-SOEs.

**Table 5.** Heterogeneity Analysis: SOEs and Non-SOEs.

| | (1) | (2) | (3) | (4) | (5) | (6) |
|---|---|---|---|---|---|---|
| | **State-Owned Enterprises** | | | **Non-State-Owned Enterprises** | | |
| | **TSUM** | **TSUM** | **TSUM** | **TSUM** | **TSUM** | **TSUM** |
| CB | 0.718 *** | | 0.083 | 0.872 *** | | −1.735 *** |
| | (3.210) | | (0.120) | (5.595) | | (−3.336) |
| Bcredit | 0.839 ** | | 3.659 *** | 2.720 *** | | 3.076 *** |
| | (2.300) | | (3.506) | (8.898) | | (3.446) |
| SUB | 0.465 *** | | −2.032 *** | 1.243 *** | | 4.021 *** |
| | (3.976) | | (−6.547) | (10.267) | | (10.527) |
| INST | 0.003 | | −1.542 *** | 0.034 | | −0.099 |
| | (0.033) | | (−3.840) | (0.477) | | (−0.296) |
| HE | 0.328 *** | | 0.370 | 0.159 ** | | 0.535 * |
| | (2.615) | | (0.758) | (2.099) | | (1.693) |
| STDI | −0.051 *** | | 0.091 *** | −0.047 *** | | 0.040 |
| | (−5.935) | | (3.387) | (−4.590) | | (1.022) |
| LTDI | 0.023 *** | | −0.097 *** | 0.013 * | | 0.049 |
| | (3.209) | | (−4.486) | (1.699) | | (1.527) |
| GOVATT | | 0.024 *** | 0.010 | | 0.004 | 0.006 |
| | | (6.678) | (1.458) | | (1.491) | (1.317) |
| GOVATT × CB | | | 0.008 | | | 0.029 *** |
| | | | (1.070) | | | (5.190) |
| GOVATT × Bcredit | | | −0.029 *** | | | −0.004 |
| | | | (−2.701) | | | (−0.372) |
| GOVATT × SUB | | | 0.025 *** | | | −0.031 *** |
| | | | (8.731) | | | (−7.638) |
| GOVATT × INST | | | 0.009 *** | | | 0.001 |
| | | | (2.591) | | | (0.499) |
| GOVATT × HE | | | 0.002 | | | −0.004 |
| | | | (0.471) | | | (−1.252) |
| GOVATT × STDI | | | −0.000 *** | | | −0.000 ** |
| | | | (−5.341) | | | (−2.293) |
| GOVATT × LTDI | | | 0.000 *** | | | −0.000 |
| | | | (5.658) | | | (−1.069) |
| _cons | 3.026 | −2.143 | −0.485 | 1.783 | −4.431 *** | 1.733 |
| | (1.345) | (−1.032) | (−0.210) | (1.021) | (−2.648) | (0.965) |
| control | yes | yes | yes | yes | yes | yes |
| industry | yes | yes | yes | yes | yes | yes |
| year | yes | yes | yes | yes | yes | yes |
| N | 9190 | 9190 | 9190 | 12,170 | 12,170 | 12,170 |
| r2 | 0.353 | 0.349 | 0.370 | 0.277 | 0.263 | 0.282 |

The *t*-statistics are in parentheses. * $p < 0.1$, ** $p < 0.05$, *** $p < 0.01$.

Second, to further analyze the differences in the effects of each component of the national green innovation system on SOEs and non-SOEs, Models 1 to 6 in Table 5 were tested for differences in coefficients between groups using Fisher's permutation test [116]. The results of Fisher's permutation test are shown in Table 6. The extent to which corporate cooperation (business innovation cooperation and good business relationships) and government subsidies positively influence green innovation in non-SOEs is greater. However, the degree of the positive impact of university-enterprise industry-research cooperation on green innovation in SOEs is much higher. Government attention has a significantly stronger effect on green innovation in SOEs. Similarly, the strengthening effect of government attention on the positive impact of corporate cooperation on corporate green innovation was greater for non-SOEs. However, the extent of the strengthening effect of government attention on the positive impact of government subsidies and industry-research cooperation on corporate green innovation is greater for SOEs. There is no significant difference in the impact of bank credit on green innovation between SOEs and non-SOEs and no significant difference in the moderating effect of government attention on the impact of bank credit on green innovation.

**Table 6.** Test for differences between groups: SOEs and Non-SOEs.

| | (1) | (2) | (1) |
|---|---|---|---|
| | Empirical *p*-Value (4)−(1) TSUM | Empirical *p*-Value (5)−(2) TSUM | Empirical *p*-Value (6)−(3) TSUM |
| CB | 0.154 * | | |
| Bcredit | 1.881 *** | | |
| SUB | 0.778 *** | | |
| INST | 0.031 | | |
| HE | −0.169 *** | | |
| STDI | 0.004 | | |
| LTDI | −0.010 | | |
| GOVATT | | −0.020 *** | |
| GOVATT × CB | | | 0.022 *** |
| GOVATT × Bcredit | | | 0.025 ** |
| GOVATT × SUB | | | −0.056 *** |
| GOVATT × INST | | | −0.007 ** |
| GOVATT × HE | | | −0.007 *** |
| GOVATT × STDI | | | 0.000 |
| GOVATT × LTDI | | | −0.000 ** |

Empirical *p*-values were used to test the significance of the differences in regression coefficients between the SOE and non-SOE in Table 5, and empirical *p*-values were obtained by bootstrap sampling 1000 times based on Fisher's permutation test. * $p < 0.1$, ** $p < 0.05$, *** $p < 0.01$.

### 4.4. Endogeneity Issues

Based on the econometric model used in empirical analyses, we must consider the endogeneity of the main explanatory variables and perform a retest after controlling for endogeneity. The main endogeneity issues are omitted variables, sample selection, and reciprocal causality. Econometric theory provides instrumental variables to address the endogeneity problem of the main explanatory variables in the econometric models. The instrumental variable is supposed to be an exogenous variable itself, theoretically uncorrelated with the explanatory variable, and related to the explanatory variable with endogeneity, which affects the explanatory variable by influencing the endogenous explanatory variable being instrumented. From a methodological perspective, two-stage least squares estimation (2SLS) using instrumental variables is a relatively mature measure [117].

The instrumental variables are the number of years a business has been established (found) and the proportion of regional R&D investment to GDP (RDINT). Good business cooperation and government subsidies are related to the number of years a company has been established. The longer a company is established, the easier it is to establish a good relationship with other companies and the government. These variables are connected to endogenous explanatory factors but not the error term, indicating that the instrumental variable is entirely exogenous. The DWH test is an important method for testing endogeneity [118]. According to the DWH test in Table 7, the explanatory variables of commercial credit (Bcredit), regional university R&D intensity (HE), and government attention (GOVATT) are exogenous, and the other variables are endogenous. The instrumental variable two-stage least-squares regression results are presented in Table 8. The coefficients remain significant in the second stage, and the findings for the weak instrumental factors are significantly greater than 10, thus rejecting the hypothesis of weak instrumental variables. The Sargan test proved the rationality of selecting instrumental variables.

To address the problem of endogeneity due to reciprocal causality, we use lagged explanatory variables for the panel data regression. The results of the lagged dependent variable regressions are listed in Table 9. The coefficients of CB, Bcredit, SUB, HE, STDI, and LTDI are consistent with the results in Table 4, indicating that our findings are valid. After adjusting for endogeneity, the major outcomes of this study remained valid.

**Table 7.** The DWH test results.

| Null Hypotheses | (1) Durbin chi2-Statistics | (2) Durbin *p*-Value | (3) Wu-Hausman F-Statistics | (4) Wu-Hausman *p*-Value |
|---|---|---|---|---|
| CB is exogenous | 13.473 | 0.000 | 13.458 | 0.000 |
| Bcredit is exogenous | 1.333 | 0.248 | 1.332 | 0.249 |
| SUB is exogenous | 12.397 | 0.000 | 12.382 | 0.000 |
| INST is exogenous | 4.097 | 0.043 | 4.090 | 0.043 |
| HE is exogenous | 0.054 | 0.816 | 0.054 | 0.816 |
| STDI is exogenous | 4.617 | 0.032 | 4.610 | 0.032 |
| LTDI is exogenous | 4.207 | 0.040 | 4.200 | 0.040 |
| GOVATT is exogenous | 1.409 | 0.235 | 1.406 | 0.236 |

**Table 8.** Instrumental variable 2-stage least squares regression.

| | (1) CB | (2) TSUM | (3) SUB | (4) TSUM | (5) INST | (6) TSUM | (7) STDI | (8) TSUM | (9) LTDI | (10) TSUM |
|---|---|---|---|---|---|---|---|---|---|---|
| found | 0.001 * (1.726) | | 0.001 * (1.836) | | −0.014 *** (−14.120) | | −0.021 * (−1.663) | | −0.048 *** (−3.107) | |
| RDINT | 0.020 *** (8.423) | | 0.032 *** (6.416) | | 0.994 *** (156.165) | | −0.029 ** (−2.447) | | 0.219 *** (2.603) | |
| CB | | 9.345 *** (3.688) | | | | | | | | |
| Bcredit | | | | 5.790 *** (3.693) | | | | | | |
| SUB | | | | | | 0.177 *** (3.839) | | | | |
| HE | | | | | | | | −1.140 * (−1.644) | | |
| STDI | | | | | | | | | | 0.332 ** (2.002) |
| _cons | −2.230 *** (−33.051) | 18.809 *** (3.274) | −4.473 *** (−24.882) | 23.898 *** (3.377) | −2.152 *** (−18.821) | −1.489 (−1.105) | −25.210 *** (−10.649) | −30.857 (−1.518) | −28.092 *** (−9.370) | 7.837 (1.577) |
| control | yes | yes | yes | yes | yes | yes | yes | yes | yes | yes |
| industry | yes | yes | yes | yes | yes | yes | yes | yes | yes | yes |
| year | yes | yes | yes | yes | yes | yes | yes | yes | yes | yes |
| First-stage F | 34.615 | | 34.214 | | 36,447.4 | | 16.331 | | 11.934 | |
| Sargan *p*-Value | 0.609 | | 0.523 | | 0.1798 | | 0.069 | | 0.150 | |
| N | 21,360 | 21,360 | 21,360 | 21,360 | 21,360 | 21,360 | 21,360 | 21,360 | 21,360 | 21,360 |
| r2 | 0.226 | 0.170 | 0.446 | 0.181 | 0.792 | 0.304 | 0.045 | 0.294 | 0.119 | 0.182 |

The *t*-statistics are in parentheses. * $p < 0.1$, ** $p < 0.05$, *** $p < 0.014$.

**Table 9.** Lagged independent variable regressions.

| | (1) L.TSUM | (2) L.TSUM | (3) L.TSUM | (4) L.TPK | (5) L.TPK | (6) L.TPK |
|---|---|---|---|---|---|---|
| CB | 0.607 *** (10.375) | | −0.275 (−1.463) | 0.159 *** (20.193) | | 0.114 *** (4.488) |
| Bcredit | 0.882 *** (8.406) | | 1.254 *** (4.108) | 0.135 *** (9.522) | | 0.168 *** (4.088) |
| SUB | 0.226 *** (6.370) | | 0.147 (1.539) | −0.003 (−0.526) | | 0.046 *** (3.535) |
| INST | −0.026 (−1.101) | | −0.132 (−1.159) | −0.015 *** (−4.761) | | 0.008 (0.545) |
| HE | 0.176 *** (5.882) | | 0.246 ** (2.000) | 0.026 *** (6.400) | | −0.001 (−0.039) |
| STDI | −0.004 * (−1.653) | | −0.017 * (−1.896) | 0.000 (1.255) | | −0.001 (−0.827) |
| LTDI | 0.009 *** (4.257) | | 0.009 (1.249) | −0.001 *** (−4.991) | | 0.001 (1.226) |

**Table 9.** *Cont.*

|  | (1) | (2) | (3) | (4) | (5) | (6) |
|---|---|---|---|---|---|---|
|  | **L.TSUM** | **L.TSUM** | **L.TSUM** | **L.TPK** | **L.TPK** | **L.TPK** |
| GOVATT |  | 0.005 *** | 0.001 |  | 0.000 ** | 0.000 ** |
|  |  | (5.156) | (0.811) |  | (2.414) | (2.004) |
| GOVATT × CB |  |  | 0.010 *** |  |  | 0.000 * |
|  |  |  | (4.914) |  |  | (1.827) |
| GOVATT × Bcredit |  |  | −0.004 * |  |  | −0.000 |
|  |  |  | (−1.818) |  |  | (−0.930) |
| GOVATT × SUB |  |  | 0.001 * |  |  | −0.000 *** |
|  |  |  | (1.852) |  |  | (−4.025) |
| GOVATT × INST |  |  | 0.001 |  |  | −0.000 * |
|  |  |  | (0.604) |  |  | (−1.759) |
| GOVATT × HE |  |  | −0.001 |  |  | 0.000 |
|  |  |  | (−0.542) |  |  | (1.470) |
| GOVATT × STDI |  |  | −0.000 ** |  |  | 0.000 |
|  |  |  | (−2.530) |  |  | (1.218) |
| GOVATT × LTDI |  |  | 0.000 *** |  |  | −0.000 *** |
|  |  |  | (2.860) |  |  | (−2.886) |
| _cons | −2.251 *** | −4.891 *** | −2.437 *** | −0.889 *** | −1.217 *** | −0.894 *** |
|  | (−3.604) | (−8.365) | (−3.772) | (−10.559) | (−15.320) | (−10.265) |
| industry | yes | yes | yes | yes | yes | yes |
| year | yes | yes | yes | yes | yes | yes |
| N | 19,224 | 19,224 | 19,224 | 19,224 | 19,224 | 19,224 |
| r2 | 0.208 | 0.197 | 0.211 | 0.197 | 0.172 | 0.199 |

The *t*-statistics are in parentheses. * $p < 0.1$, ** $p < 0.05$, *** $p < 0.01$.

### 4.5. Robustness Checks

An invention patent is a new technical solution proposed by the patent administrative department of the State Council for a product, method, or improvement that is granted to the owner of the solution for a certain period of exclusive implementation after substantive and preliminary examinations. Invention patents are more difficult to apply for, have higher technical value, and have longer protection times for enterprises. Therefore, green invention patents are of high research value in the study of corporate green innovation, and this study uses green invention patent output and patent diversity as explanatory variables to conduct robustness tests. The findings in Table 10 suggest that the coefficients are statistically significant, consistent with prior findings.

**Table 10.** Robustness checks.

|  | (1) | (2) | (3) | (4) | (5) | (6) |
|---|---|---|---|---|---|---|
|  | **ISUM** | **ISUM** | **ISUM** | **IPK** | **IPK** | **IPK** |
| CB | 0.251 *** |  | −0.149 | 0.117 *** |  | 0.057 *** |
|  | (6.418) |  | (−1.184) | (20.689) |  | (3.116) |
| Bcredit | 0.372 *** |  | 0.660 *** | 0.070 *** |  | 0.097 *** |
|  | (5.360) |  | (3.283) | (6.961) |  | (3.347) |
| SUB | 0.256 *** |  | 0.097 | 0.018 *** |  | 0.054 *** |
|  | (10.572) |  | (1.477) | (5.058) |  | (5.625) |
| INST | −0.004 |  | −0.321 *** | −0.006 *** |  | 0.005 |
|  | (−0.258) |  | (−4.282) | (−2.754) |  | (0.441) |
| HE | 0.061 *** |  | 0.064 | 0.019 *** |  | −0.006 |
|  | (3.079) |  | (0.795) | (6.521) |  | (−0.493) |
| STDI | −0.016 *** |  | 0.028 *** | −0.001 ** |  | −0.002 * |
|  | (−8.496) |  | (4.565) | (−2.237) |  | (−1.848) |

**Table 10.** *Cont.*

|  | (1) | (2) | (3) | (4) | (5) | (6) |
|---|---|---|---|---|---|---|
|  | ISUM | ISUM | ISUM | IPK | IPK | IPK |
| LTDI | 0.005 *** | | −0.006 | −0.001 *** | | −0.000 |
| | (3.104) | | (−1.221) | (−6.019) | | (−0.577) |
| GOVATT | | 0.004 *** | 0.001 | | 0.000 *** | 0.000 |
| | | (5.495) | (0.747) | | (4.544) | (1.230) |
| GOVATT × CB | | | 0.005 *** | | | 0.001 *** |
| | | | (3.402) | | | (3.437) |
| GOVATT × Bcredit | | | −0.003 * | | | −0.000 |
| | | | (−1.688) | | | (−1.060) |
| GOVATT × SUB | | | 0.002 *** | | | −0.000 *** |
| | | | (2.601) | | | (−4.075) |
| GOVATT × INST | | | 0.002 *** | | | −0.000 |
| | | | (3.375) | | | (−1.435) |
| GOVATT × HE | | | 0.000 | | | 0.000 ** |
| | | | (0.463) | | | (2.092) |
| GOVATT × STDI | | | −0.000 *** | | | 0.000 |
| | | | (−7.390) | | | (1.189) |
| GOVATT × LTDI | | | 0.000 ** | | | −0.000 |
| | | | (2.307) | | | (−1.345) |
| _cons | 3.063 *** | 1.343 *** | 2.891 *** | −0.513 *** | −0.810 *** | −0.513 *** |
| | (7.528) | (3.525) | (6.879) | (−8.726) | (−14.618) | (−8.429) |
| industry | yes | yes | yes | yes | yes | yes |
| year | yes | yes | yes | yes | yes | yes |
| N | 21,360 | 21,360 | 21,360 | 21,360 | 21,360 | 21,360 |
| r2 | 0.303 | 0.295 | 0.308 | 0.233 | 0.211 | 0.234 |

The *t*-statistics are in parentheses. * $p < 0.1$, ** $p < 0.05$, *** $p < 0.01$.

To verify the reliability of the model, this study uses a series model for robustness testing, a generalized linear model, and a firm and time fixed-effects model for regression. In the framework of the generalized linear model, dependent variables no longer need to be continuous and normal, and independent variables are less specific [119]. The fixed effects model addresses the problem of omitted variables in the endogeneity problem and increases regression robustness using the Rogers standard error. Rogers' standard error estimators are robust to disturbances that are heteroscedastic and autocorrelated [120]. The regression results are presented in Table 11.

**Table 11.** Model replacement regression.

|  | (1) | (2) | (3) | (4) | (5) | (6) |
|---|---|---|---|---|---|---|
|  | | **Rogers SE** | | | **GLM** | |
|  | TSUM | TSUM | TSUM | TSUM | TSUM | TSUM |
| CB | 0.287 * | | −0.459 | 0.843 *** | | −0.722 * |
| | (1.957) | | (−0.738) | (6.613) | | (−1.709) |
| Bcredit | 0.909 *** | | 3.150 *** | 1.321 *** | | 3.398 *** |
| | (2.626) | | (2.602) | (6.196) | | (5.034) |
| SUB | 0.702 * | | 0.017 | 0.747 *** | | −0.300 |
| | (1.914) | | (0.017) | (9.179) | | (−1.355) |
| INST | −0.167 | | −1.910 ** | −0.005 | | −1.319 *** |
| | (−0.192) | | (−1.975) | (−0.088) | | (−5.241) |

**Table 11.** *Cont.*

|  | (1) | (2) | (3) | (4) | (5) | (6) |
|---|---|---|---|---|---|---|
|  |  | **Rogers SE** |  |  | **GLM** |  |
|  | **TSUM** | **TSUM** | **TSUM** | **TSUM** | **TSUM** | **TSUM** |
| HE | 0.253 |  | 1.682 *** | 0.195 *** |  | 0.521 * |
|  | (0.778) |  | (4.102) | (2.934) |  | (1.922) |
| STDI | −0.044 ** |  | 0.118 | −0.049 *** |  | 0.092 *** |
|  | (−2.529) |  | (1.434) | (−7.816) |  | (4.434) |
| LTDI | 0.017 ** |  | −0.087 | −0.021 *** |  | −0.066 *** |
|  | (1.986) |  | (−1.385) | (−4.155) |  | (−4.097) |
| GOVATT |  | 0.011 *** | 0.011 ** |  | 0.013 *** | 0.006 |
|  |  | (3.107) | (2.179) |  | (6.083) | (1.494) |
| GOVATT × CB |  |  | 0.009 * |  |  | 0.018 *** |
|  |  |  | (1.739) |  |  | (3.999) |
| GOVATT × Bcredit |  |  | −0.024 * |  |  | −0.023 *** |
|  |  |  | (−1.694) |  |  | (−3.187) |
| GOVATT × SUB |  |  | 0.007 * |  |  | 0.011 *** |
|  |  |  | (1.690) |  |  | (5.077) |
| GOVATT × INST |  |  | 0.010 *** |  |  | 0.009 *** |
|  |  |  | (3.008) |  |  | (4.309) |
| GOVATT × HE |  |  | −0.005 |  |  | −0.002 |
|  |  |  | (−1.420) |  |  | (−0.661) |
| GOVATT × STDI |  |  | −0.000 * |  |  | −0.000 *** |
|  |  |  | (−1.765) |  |  | (−7.006) |
| GOVATT × LTDI |  |  | 0.000 * |  |  | 0.000 *** |
|  |  |  | (1.812) |  |  | (5.581) |
| _cons | 3.445 | −0.178 | 0.826 | 1.152 | −4.308 *** | 0.133 |
|  | (0.582) | (−0.031) | (0.151) | (0.892) | (−3.564) | (0.099) |
| firm | yes | yes | yes | yes | yes | yes |
| year | yes | yes | yes | yes | yes | yes |
| N | 21,360 | 21,360 | 21,360 | 21,360 | 21,360 | 21,360 |
| r2 | 0.468 | 0.466 | 0.474 | 0.468 | 0.466 | 0.474 |

The *t*-statistics are in parentheses. * $p < 0.1$, ** $p < 0.05$, *** $p < 0.01$.

## 5. Discussion

### 5.1. Theoretical Contribution

The main theoretical contributions of this study are as follows: First, it proposes a national green innovation system based on RDT and NIS. This study examines an innovation ecosystem comprising the government, research institutions (universities), enterprises, and financial institutions to increase corporate green innovation and promote national ecological development. This study adds to the resource dependence theory in the field of green innovation by focusing on the impact of specific resources of each participant in the green innovation system on corporate green innovation. This study obtained consistent findings with existing studies on factors that increase green innovation such as policy regulation [15,121], government subsidies [89,118], financing constraints [18], industry-academia collaboration [23], and corporate collaboration [75,77]. With the introduction of the concept of government attention, this study finds the government is able to effectively link various players in the green innovation system. Government attention is found to be an effective approach to coordinate collaborative innovation among participants in a green innovation system, providing a new line of research for resource dependence theory and green innovation. In addition, this study introduces financial institutions into the green innovation system to fill the gap of neglecting the complementarity between NIS and financial institutions in the study of green innovation system [30].

Second, to analyze the role of the government in national green innovation systems, this study introduces the concept of attention. The prefecture-level municipal government work report is both a summary of the government's work in the past year and a disclosure

of the government's priorities in the coming years. Therefore, this study measures government attention based on the number of occurrences of words and phrases related to ecological development, green innovation, and innovation talent concerns in the annual work reports of prefecture-level municipal governments in China through textual analysis of these reports. Through an empirical analysis of the national green innovation system, this study finds that government attention has a moderating role in the effects of enterprise cooperation, industry-academia cooperation, financial institutions, and government subsidies on the output of green innovation and the patent diversity of enterprises.

Third, this study constructs a Nash-Cournot equilibrium under the national green innovation system, incorporating the effects of corporate cooperation, industry-research cooperation, government subsidies, policy regulation, and bank loans. The results show that when external green R&D funds such as loans and subsidies of enterprises are higher than internal green R&D investment, the economic effect of the market equilibrium achieved by corporate green innovation is better than that of the market equilibrium without green innovation. Furthermore, industry-research cooperation is beneficial for enterprise green innovation. This study extends the research on corporate cooperation on corporate innovation [106], government subsidies, and loan size on green innovation [16]. A positive effect of the national green innovation system constructed in this study on green enterprise innovation was also found.

### 5.2. Practice Implication

This study proposes a national green innovation system and finds positive effects of firms' innovation cooperation experience, good commercial credit, government subsidies, and regional university R&D intensity on firms' green innovation output and green innovation patent diversity. In a study on the impact of bank loans on corporate green innovation, differences were found between the impact of short-term and long-term loans on corporate green innovation. Short-term loans are unfavorable to corporate green innovation output and green innovation patent diversity; long-term loans are favorable to corporate green innovation output but unfavorable to green innovation patent diversity. Government attention has a direct positive impact on corporate green innovation and strengthens the positive impact of collaborative innovation experience, government subsidies, and long-term loans on corporate green innovation output. This study found that SOEs have significantly higher green patent output, patent diversity, government attention, corporate cooperation, government subsidies, and bank credit than their non-SOE counterparts. However, the effects of corporate collaboration and government subsidies on green innovation are significantly higher in non-SOEs than in SOEs, and the intensity of the moderating role of government attention on the effects of non-SOE collaboration on green innovation is higher in SOEs. The implications of the study's results are presented below.

This study finds the role of government, banks, research institutions (universities), and industry collaboration in enhancing corporate green innovation with Chinese firms as the research sample. This provides practical implications for countries that are committed to increasing corporate green innovation for national ecological development. First, at the government level, it is essential to clarify the government's leadership position in the national green innovation system and ensure continuous attention to green innovation development. The state should ensure green innovation intellectual property protection, encourage enterprises to cooperate in green innovation, promote industry-academia green innovation, improve enterprise credit processes, and optimize enterprise loan structures. Second, at the bank level, there is a need to accelerate the approval process for enterprises to obtain loans, improve the structure of enterprise loans, and optimize the interest rate of enterprise loans. Long-term corporate loans should be the primary financing method for high-quality enterprises, reducing corporate loan fees and encouraging innovation. Third, at the enterprise level, enterprises should leverage green innovation R&D cooperation, create a suitable operational environment for industry cooperation, and actively cooperate with universities and research institutions for green innovation. Finally, from

the perspective of research institutions and universities, the government should encourage intellectual exchange between enterprises and research institutions, broaden the channels for knowledge exchange between industry and academia, and ensure that the outcomes of research institutions and universities on green innovation can be exchanged to benefit enterprises and green development in the country. To increase the development of national green innovation, it is necessary to establish and improve the national green innovation system; recognize the collaboration among the government, enterprises, R&D institutions, and financial institutions to improve the legislative protection of intellectual property rights; encourage green innovation cooperation among enterprises; take advantage of financial institutions' financing services; and improve the cooperation path between enterprises and universities.

### 5.3. Limitations and Future Research

Green innovation is an emerging concept in China, and the disclosure system for corporate green innovation is not well developed, despite the country's commitment to environmental protection. With improvements in legislation related to corporate environmental information, more scholars will focus on corporate green innovation research.

Second, the national green innovation system requires the participation of government, financial institutions, R&D institutions, and enterprises. However, currently, only data from provincial research institutes and universities are available, and such data are not sufficiently accurate. With the disclosure of research data from research institutions and universities at the prefectural level, more scholars will examine direct collaborative innovation between research institutions and firms in detail.

Third, there is little research on government attention, and no widely accepted measure of government attention. In addition to the government work reports issued by the Chinese government, some countries and economies such as the United States, Russia, and the European Union also have the state of the nation addresses [122,123], which provide scholars with an adequate research base to study government attention. Therefore, this study suggests future research may further examine the concept of government attention and its related research. In addition, this study emphasizes the important role of government influence on green innovation in firms, but scholars disagree on the role of government on green innovation [124], so future research can test the influence of government on green innovation in different countries and economies. Finally, this study proposes the concept of green innovation ecosystem and verifies its validity in China, but whether green innovation ecosystem is used in other countries or economies still needs to be tested.

## 6. Conclusions

Based on NIS and RDT, this study examines the concept of a national green innovation system. To increase green innovation and ecological development, a country should fully develop a synergy of innovation among the government, enterprises, research institutions, and financial institutions. This study provides both theoretical and empirical evidence for the positive impact of national green innovation systems on corporate green innovation. Based on the Nash-Cournot equilibrium theory, this study considers the effects of corporate R&D cooperation, government policies and subsidies, industry-academia cooperation, and bank credit on green innovation. The results show that when external R&D funding, such as loans and subsidies, is greater than internal R&D investment, the profits and equilibrium prices of firms under market equilibrium conditions are better than those without green innovation. In the empirical testing stage, this study finds that (1) good commercial credit and corporate innovation cooperation drive corporate green innovation, (2) government subsidies increase corporate green innovation, (3) regional universities' R&D intensity increases corporate green innovation, but regional R&D intensity cannot promote corporate green innovation output or even inhibit corporate green innovation patent diversity, (4) short-term borrowing is not conducive to corporate green innovation, but long-term loans are beneficial to corporate green innovation, and (5) government

attention is beneficial to corporate green innovation. This strengthens corporate cooperation and the positive influence of government subsidies on corporate green innovation. This also deepens the influence of bank credit on corporate green innovation. In the study of SOEs and non-SOEs, this study finds that although SOEs have higher green innovation output and patent diversity than non-SOEs, the positive effects of corporate collaboration and government subsidies on corporate green innovation are significantly higher in non-SOEs than those in SOEs. Moreover, the moderating effect of government attention on the impact of enterprise cooperation on green innovation was stronger than that of SOEs. Additionally, this result holds when endogeneity and robustness are considered.

**Author Contributions:** D.G., methodology, data curation, formal analysis, and writing— original draft; C.W.Y.W., conceptualization, methodology, writing—review, and editing. K.-h.L. wrote, reviewed, edited, and administered the project. All authors have read and agreed to the published version of the manuscript.

**Funding:** This research was partially supported by The Hong Kong Polytechnic University under grant number 1-ZEZY and by the Research Grants Council of the Hong Kong Special Administrative Region, China (GRF 15503222).

**Data Availability Statement:** Data are available on request owing to restrictions such as privacy or ethics.

**Conflicts of Interest:** The authors declare no conflict of interest.

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
