# Peer review of "Development of Ecosystem for Corporate Green Innovation: Resource Dependency Theory Perspective"

_sustainability, doi:10.3390/su15065450_

Round 1

Reviewer 1 Report

Some terms such as the ecosystem and the national green innovation system need to be defined. As a matter of fact, the national green innovation system seems a quite loose term. It is more like a collaborative structure getting the enterprises, educational institutions and the govenrment connected. If so, the authors need to elaborate on this term through the institutional economics persective. The government's role in this system can be controversial as the liberal econoimcs do not recognize such things that the government plays a decisive role within the institutional structure regardless of its the key source of disputes between the US and China in the ongoing trade war. The arguments in this regard can be further improved and justified. The literature review should be further improved bringing in more materials and relevant arguments. Can the conclusion be generalized to other countries/jurisdictions? If not, the value of the article is not much.

Author Response

Reviewer #1’s comment #1:

Some terms such as the ecosystem and the national green innovation system need to be defined. As a matter of fact, the national green innovation system seems a quite loose term. It is more like a collaborative structure getting the enterprises, educational institutions and the government connected. If so, the authors need to elaborate on this term through the institutional economics perspective. The government's role in this system can be controversial as the liberal economics do not recognize such things that the government plays a decisive role within the institutional structure regardless of its the key source of disputes between the US and China in the ongoing trade war. The arguments in this regard can be further improved and justified. The literature review should be further improved bringing in more materials and relevant arguments. Can the conclusion be generalized to other countries/jurisdictions? If not, the value of the article is not much.

Response to Reviewer #1’s comment #1:

In the manuscript, we define the green innovation ecosystem in Line 68-71. Green innovation ecosystem is considered the entity comprising government, research institutions (universities), enterprises, and financial institutions to increase corporate green innovation for national ecological development. In this innovation ecosystem, government provides subsidies for enterprises' green innovation; banks provide financial support for green innovation; enterprises cooperate with research institutions in industry-academia-research to provide technical support and resource allocation for green innovation. Furthermore, government provide guidance to enterprises, research institutions, and financial institutions through resource allocation [33] and signaling effects [34].

This study considers the government plays two roles in green innovation ecosystem: first, the government provides green innovation subsidies for enterprises, and second, the government introduces policies and regulations to guide and regulate for corporate green innovation. Prior studies have found the empirical evidence of the impact of environmental regulations [13, 14], and government subsidies [15] on green innovation based on the Porter's hypothesis. The Porter’s hypothesis argues that environmental regulations promote green innovation [13]. Empirical research on Porter's hypothesis confirms the positive role of government on green innovation in China and the US as well as in other countries or economies [14]. In addition, considering large upfront investment in green innovation [30], high uncertainty of output [31], and double externality[32] it is difficult for firms to reach an optimal level of green innovation, and thus firms will be less motivated to green innovation without government support [33]. Overall, the role of government in the green innovation ecosystem is crucial.

This study found that corporate collaboration, government subsidies, regional university R&D intensity, long-term credit, and government attention enhance number of green innovation patents and patent diversity. Government attention positively strengthens the positive effects of corporate R&D cooperation, government subsidies, R&D intensity of regional research institutions, and long-term credit on number of green innovation patents. Such findings have an important role for countries committed to environmental protection. To promote corporate green innovation to achieve social sustainable development, government environmental attention in both developing and developed countries has an indelible role. In Section 5.2 (Line 688-691), we highlight that this study provides some practice implications for countries that are committed to increasing corporate green innovation to achieve national eco-development by using Chinese listed enterprises as an example for empirical research.

  1. Peng, H.; Shen, N.; Ying, H.; Wang, Q., Can environmental regulation directly promote green innovation behavior?——based on situation of industrial agglomeration. Journal of Cleaner Production 2021, 314, 128044
  2. Zhang, D.; Zheng, M.; Feng, G.-F.; Chang, C.-P., Does an environmental policy bring to green innovation in renewable energy? Renewable Energy 2022, 195, 1113-1124
  3. Huang, Z.; Liao, G.; Li, Z., Loaning scale and government subsidy for promoting green innovation. Technological Forecasting and Social Change 2019, 144, 148-156
  4. Yang, Q.-C.; Zheng, M.; Chang, C.-P., Energy policy and green innovation: A quantile investigation into renewable energy. Renewable Energy 2022, 189, 1166-1175
  5. Bi, G.; Jin, M.; Ling, L.; Yang, F., Environmental subsidy and the choice of green technology in the presence of green consumers. Annals of Operations Research 2017, 255, (1), 547-568
  6. Qi, G.; Zou, H.; Xie, X., Governmental inspection and green innovation: Examining the role of environmental capability and institutional development. Corporate Social Responsibility and Environmental Management 2020, 27, (4), 1774-1785
  7. Bai, Y.; Song, S.; Jiao, J.; Yang, R., The impacts of government R&D subsidies on green innovation: Evidence from Chinese energy-intensive firms. Journal of Cleaner Production 2019, 233, 819-829
  8. Wang, Z.; Li, X.; Xue, X.; Liu, Y., More government subsidies, more green innovation? The evidence from Chinese new energy vehicle enterprises. Renewable Energy 2022, 197, 11-21

We have revised our manuscript according to your comments and suggestions.

Reviewer 2 Report

1- line:48-49 problem statement is clear. However, it lacks of supporting and evidence from LR  or industrial deports.  Please add  some  supporting and evidence.

2- Line 68: "In this innovation ecosystem " why you add word "this" ? do you refer to a certain innovation ecosystem.   

3- Figure 1. looks Good and innovative. However, the directional  flow of some Resources specially (Financial support & Technical resources)  should be bidirectional <______> not single. direction. to confirm the two ways dynamism of resource sharing flow and transfer from and to  Enterprises and other parities.   

4- Table 10: Should be place in the next new page. 

5- Even though analysis is rigor , too long , involve many codes and regressions, a simple reader cannot find clear simple section about { discussion on developed hypothesis}. Authors, have developed list of hypothesis and done long complex process analysis. That should ends with clear section on  which  hypothesis were supported? which one were rejected and why ? that was missing or not really clear.   

Author Response

Reviewer #2’s comment #1:

Line:48-49 problem statement is clear. However, it lacks supporting and evidence from LR or industrial deports. Please add some supporting and evidence.

Response to Reviewer #2’s comment #1:

Based on Reviewer #2’s suggestions, we add relevant literature support and evidence.

In Line 48-52, we revise the text by adding more literature support.

“Previous scholars studies on green innovation ecosystem include the game between enterprises and upstream and downstream enterprises [23, 24], the government-university-industry cooperative alliance [22],and the interaction mechanism between external environmental regulation and corporate internal green innovation process [25].”

  1. Yang, Z.; Chen, H.; Du, L.; Lin, C.; Lu, W., How does alliance-based government-university-industry foster cleantech innovation in a green innovation ecosystem? Journal of Cleaner Production 2021, 283, 124559
  2. Zou, H.; Qin, H.; He, D.; Sun, J., Research on an enterprise green innovation ecosystem from the vulnerability perspective: evolutionary game and simulation. Ieee Access 2021, 9, 140809-140823
  3. Zeng, W.; Li, L.; Huang, Y., Industrial collaborative agglomeration, marketization, and green innovation: Evidence from China’s provincial panel data. Journal of Cleaner Production 2021, 279, 123598
  4. Zeng, J.; Chen, X.; Liu, Y.; Cui, R.; Zhao, P., How does the enterprise green innovation ecosystem collaborative evolve? Evidence from China. Journal of Cleaner Production 2022, 375, 134181

Reviewer #2’s comment #2:

Line 68: "In this innovation ecosystem " why you add word "this”? do you refer to a certain innovation ecosystem.  

Response to Reviewer #2’s comment #2:

Compared to the green innovation ecosystem proposed by previous scholars, this study considers the influences of government attention and financial institutions. Specifically, the green innovation ecosystem is considered as entity consisting of government, research institutions (universities), enterprises, and financial institutions to increase corporate green innovation for national ecological development. We refer specifically to this innovative green innovation system that we propose in Line 63-67.

Reviewer #2’s comment #3:

Figure 1. looks Good and innovative. However, the directional flow of some Resources specially (Financial support & Technical resources) should be bidirectional no single direction. To confirm the two ways dynamism of resource sharing flow and transfer from and to enterprises and other parties.  

Response to Reviewer #2’s comment #3:

We greatly appreciate your comments, and we edit Figure 1 based on your advice. We change the flow direction of financial support and technical resources to a two-way direction.

Figure 1. National Green Innovation System

Reviewer #2’s comment #4:

Table 10: Should be place in the next new page.

Response to Reviewer #2’s comment #4:

We place Table 10 and Table 11 on new pages.

Reviewer #2’s comment #5:

Even though analysis is rigor, too long, involve many codes and regressions, a simple reader cannot find clear simple section about {discussion on developed hypothesis}. Authors have developed list of hypotheses and done long complex process analysis. That should end with clear section on which hypothesis were supported? which one were rejected and why? that was missing or not clear.   

Response to Reviewer #2’s comment #5:

Thanks for the comments. We add the following discussion and Table 4b in Section 4.2 with aim to summarize and provide an overview of the results:

“The results of the hypotheses are summarized in Table 4b. Specifically, corporate collaboration, government subsidies, regional research institutions research intensity, and long-term loans are positively associated with corporate green innovation, while short-term loans are negatively associated with corporate green innovation, i.e., hypotheses 1a, 1b, 2, 3a, 3b, 4a, and 4b are supported. Government attention strengthens the impact of corporate collaborative innovation, government subsidies, research institutions and bank loans on the number of corporate green patents. Government attention also strengthens the impact of corporate collaborative innovation on corporate green patent diversity. Government attention weakens the impact of corporate commercial credit, government subsidies, regional research institutions' research intensity and long-term loans on corporate green patent diversity.

Table 4b. The results of the hypothesis 1a to hypothesis 5f.

Hypotheses

Results

1a: Enterprises with granted patents jointly with other enterprises or institution are likely to achieve corporate green innovation.

Support

1b: Corporate commercial credit is positively associated with corporate green innovation.

Support

2: Government subsidies are positively associated with corporate green innovation.

Support

3a: Regional university research intensity is positively associated with corporate green innovation.

No

3b: Regional research institutions research intensity is positively associated with corporate green innovation.

Support

4a: Short-term loans are negatively associated with corporate green innovation.

Support

4b: Long-term loans are positively associated with corporate green innovation.

Support

5a: Government attention strengthens the impact of corporate collaboration innovation on corporate green innovation.

Support

5b: Government attention strengthens the impact of corporate commercial credit on corporate green innovation.

No

5c: Government attention strengthens the impact of government subsidies on corporate green innovation.

Partially

5d: Government attention strengthens the impact of R&D intensity of regional universities on corporate green innovation.

No

5e: Government attention strengthens the impact of R&D intensity of research institutions on corporate green innovation.

Partially

5f: Government attention strengthens the impact of bank loans on corporate green innovation.

Partially

Reviewer #2’s comment #1:

Line:48-49 problem statement is clear. However, it lacks supporting and evidence from LR or industrial deports. Please add some supporting and evidence.

Response to Reviewer #2’s comment #1:

Based on Reviewer #2’s suggestions, we add relevant literature support and evidence.

In Line 48-52, we revise the text by adding more literature support.

“Previous scholars studies on green innovation ecosystem include the game between enterprises and upstream and downstream enterprises [23, 24], the government-university-industry cooperative alliance [22],and the interaction mechanism between external environmental regulation and corporate internal green innovation process [25].”

  1. Yang, Z.; Chen, H.; Du, L.; Lin, C.; Lu, W., How does alliance-based government-university-industry foster cleantech innovation in a green innovation ecosystem? Journal of Cleaner Production 2021, 283, 124559
  2. Zou, H.; Qin, H.; He, D.; Sun, J., Research on an enterprise green innovation ecosystem from the vulnerability perspective: evolutionary game and simulation. Ieee Access 2021, 9, 140809-140823
  3. Zeng, W.; Li, L.; Huang, Y., Industrial collaborative agglomeration, marketization, and green innovation: Evidence from China’s provincial panel data. Journal of Cleaner Production 2021, 279, 123598
  4. Zeng, J.; Chen, X.; Liu, Y.; Cui, R.; Zhao, P., How does the enterprise green innovation ecosystem collaborative evolve? Evidence from China. Journal of Cleaner Production 2022, 375, 134181

Reviewer #2’s comment #2:

Line 68: "In this innovation ecosystem " why you add word "this”? do you refer to a certain innovation ecosystem.  

Response to Reviewer #2’s comment #2:

Compared to the green innovation ecosystem proposed by previous scholars, this study considers the influences of government attention and financial institutions. Specifically, the green innovation ecosystem is considered as entity consisting of government, research institutions (universities), enterprises, and financial institutions to increase corporate green innovation for national ecological development. We refer specifically to this innovative green innovation system that we propose in Line 63-67.

Reviewer #2’s comment #3:

Figure 1. looks Good and innovative. However, the directional flow of some Resources specially (Financial support & Technical resources) should be bidirectional no single direction. To confirm the two ways dynamism of resource sharing flow and transfer from and to enterprises and other parties.  

Response to Reviewer #2’s comment #3:

We greatly appreciate your comments, and we edit Figure 1 based on your advice. We change the flow direction of financial support and technical resources to a two-way direction.

Figure 1. National Green Innovation System

Reviewer #2’s comment #4:

Table 10: Should be place in the next new page.

Response to Reviewer #2’s comment #4:

We place Table 10 and Table 11 on new pages.

Reviewer #2’s comment #5:

Even though analysis is rigor, too long, involve many codes and regressions, a simple reader cannot find clear simple section about {discussion on developed hypothesis}. Authors have developed list of hypotheses and done long complex process analysis. That should end with clear section on which hypothesis were supported? which one were rejected and why? that was missing or not clear.   

Response to Reviewer #2’s comment #5:

Thanks for the comments. We add the following discussion and Table 4b in Section 4.2 with aim to summarize and provide an overview of the results:

“The results of the hypotheses are summarized in Table 4b. Specifically, corporate collaboration, government subsidies, regional research institutions research intensity, and long-term loans are positively associated with corporate green innovation, while short-term loans are negatively associated with corporate green innovation, i.e., hypotheses 1a, 1b, 2, 3a, 3b, 4a, and 4b are supported. Government attention strengthens the impact of corporate collaborative innovation, government subsidies, research institutions and bank loans on the number of corporate green patents. Government attention also strengthens the impact of corporate collaborative innovation on corporate green patent diversity. Government attention weakens the impact of corporate commercial credit, government subsidies, regional research institutions' research intensity and long-term loans on corporate green patent diversity.

Table 4b. The results of the hypothesis 1a to hypothesis 5f.

Hypotheses

Results

1a: Enterprises with granted patents jointly with other enterprises or institution are likely to achieve corporate green innovation.

Support

1b: Corporate commercial credit is positively associated with corporate green innovation.

Support

2: Government subsidies are positively associated with corporate green innovation.

Support

3a: Regional university research intensity is positively associated with corporate green innovation.

No

3b: Regional research institutions research intensity is positively associated with corporate green innovation.

Support

4a: Short-term loans are negatively associated with corporate green innovation.

Support

4b: Long-term loans are positively associated with corporate green innovation.

Support

5a: Government attention strengthens the impact of corporate collaboration innovation on corporate green innovation.

Support

5b: Government attention strengthens the impact of corporate commercial credit on corporate green innovation.

No

5c: Government attention strengthens the impact of government subsidies on corporate green innovation.

Partially

5d: Government attention strengthens the impact of R&D intensity of regional universities on corporate green innovation.

No

5e: Government attention strengthens the impact of R&D intensity of research institutions on corporate green innovation.

Partially

5f: Government attention strengthens the impact of bank loans on corporate green innovation.

Partially

Reviewer #2’s comment #1:

Line:48-49 problem statement is clear. However, it lacks supporting and evidence from LR or industrial deports. Please add some supporting and evidence.

Response to Reviewer #2’s comment #1:

Based on Reviewer #2’s suggestions, we add relevant literature support and evidence.

In Line 48-52, we revise the text by adding more literature support.

“Previous scholars studies on green innovation ecosystem include the game between enterprises and upstream and downstream enterprises [23, 24], the government-university-industry cooperative alliance [22],and the interaction mechanism between external environmental regulation and corporate internal green innovation process [25].”

  1. Yang, Z.; Chen, H.; Du, L.; Lin, C.; Lu, W., How does alliance-based government-university-industry foster cleantech innovation in a green innovation ecosystem? Journal of Cleaner Production 2021, 283, 124559
  2. Zou, H.; Qin, H.; He, D.; Sun, J., Research on an enterprise green innovation ecosystem from the vulnerability perspective: evolutionary game and simulation. Ieee Access 2021, 9, 140809-140823
  3. Zeng, W.; Li, L.; Huang, Y., Industrial collaborative agglomeration, marketization, and green innovation: Evidence from China’s provincial panel data. Journal of Cleaner Production 2021, 279, 123598
  4. Zeng, J.; Chen, X.; Liu, Y.; Cui, R.; Zhao, P., How does the enterprise green innovation ecosystem collaborative evolve? Evidence from China. Journal of Cleaner Production 2022, 375, 134181

Reviewer #2’s comment #2:

Line 68: "In this innovation ecosystem " why you add word "this”? do you refer to a certain innovation ecosystem.  

Response to Reviewer #2’s comment #2:

Compared to the green innovation ecosystem proposed by previous scholars, this study considers the influences of government attention and financial institutions. Specifically, the green innovation ecosystem is considered as entity consisting of government, research institutions (universities), enterprises, and financial institutions to increase corporate green innovation for national ecological development. We refer specifically to this innovative green innovation system that we propose in Line 63-67.

Reviewer #2’s comment #3:

Figure 1. looks Good and innovative. However, the directional flow of some Resources specially (Financial support & Technical resources) should be bidirectional no single direction. To confirm the two ways dynamism of resource sharing flow and transfer from and to enterprises and other parties.  

Response to Reviewer #2’s comment #3:

We greatly appreciate your comments, and we edit Figure 1 based on your advice. We change the flow direction of financial support and technical resources to a two-way direction.

Figure 1. National Green Innovation System

Reviewer #2’s comment #4:

Table 10: Should be place in the next new page.

Response to Reviewer #2’s comment #4:

We place Table 10 and Table 11 on new pages.

Reviewer #2’s comment #5:

Even though analysis is rigor, too long, involve many codes and regressions, a simple reader cannot find clear simple section about {discussion on developed hypothesis}. Authors have developed list of hypotheses and done long complex process analysis. That should end with clear section on which hypothesis were supported? which one were rejected and why? that was missing or not clear.   

Response to Reviewer #2’s comment #5:

Thanks for the comments. We add the following discussion and Table 4b in Section 4.2 with aim to summarize and provide an overview of the results:

“The results of the hypotheses are summarized in Table 4b. Specifically, corporate collaboration, government subsidies, regional research institutions research intensity, and long-term loans are positively associated with corporate green innovation, while short-term loans are negatively associated with corporate green innovation, i.e., hypotheses 1a, 1b, 2, 3a, 3b, 4a, and 4b are supported. Government attention strengthens the impact of corporate collaborative innovation, government subsidies, research institutions and bank loans on the number of corporate green patents. Government attention also strengthens the impact of corporate collaborative innovation on corporate green patent diversity. Government attention weakens the impact of corporate commercial credit, government subsidies, regional research institutions' research intensity and long-term loans on corporate green patent diversity.

Table 4b. The results of the hypothesis 1a to hypothesis 5f.

Hypotheses

Results

1a: Enterprises with granted patents jointly with other enterprises or institution are likely to achieve corporate green innovation.

Support

1b: Corporate commercial credit is positively associated with corporate green innovation.

Support

2: Government subsidies are positively associated with corporate green innovation.

Support

3a: Regional university research intensity is positively associated with corporate green innovation.

No

3b: Regional research institutions research intensity is positively associated with corporate green innovation.

Support

4a: Short-term loans are negatively associated with corporate green innovation.

Support

4b: Long-term loans are positively associated with corporate green innovation.

Support

5a: Government attention strengthens the impact of corporate collaboration innovation on corporate green innovation.

Support

5b: Government attention strengthens the impact of corporate commercial credit on corporate green innovation.

No

5c: Government attention strengthens the impact of government subsidies on corporate green innovation.

Partially

5d: Government attention strengthens the impact of R&D intensity of regional universities on corporate green innovation.

No

5e: Government attention strengthens the impact of R&D intensity of research institutions on corporate green innovation.

Partially

5f: Government attention strengthens the impact of bank loans on corporate green innovation.

Partially

Reviewer 3 Report

Dear authors, this paper is very excelence.

Your study found that corporate collaboration, government subsidies, regional university R&D intensity, long-term credit, and government attention enhance number of green innovation patents and patent diversity. Government attention positively strengthens the positive effects of corporate R&D cooperation, government subsidies, R&D intensity of regional research institutions, and long-term credit on number of 21 green innovation patents.

Author Response

Reviewer #3

Reviewer #3’s comment #1:

Dear authors, this paper is very excellence.

Your study found that corporate collaboration, government subsidies, regional university R&D intensity, long-term credit, and government attention enhance number of green innovation patents and patent diversity. Government attention positively strengthens the positive effects of corporate R&D cooperation, government subsidies, R&D intensity of regional research institutions, and long-term credit on number of green innovation patents.

Response to Reviewer #3’s comment #1:

Thank you very much for your time. To maintain scientific rigor and increase the readability of the manuscript, we have added Table 4b for easy reference of findings.  

Round 2

Reviewer 1 Report

The literature review only covers Chinese scholarship. There is a lack of English source of materials. Please add.

Author Response

Thank you very much for your comments. We have reorganized the literature cited in the literature review and added the following references. In addition, we polished the manuscript to ensure the readability.

  1. Clausen, L. P. W.; Nielsen, M. B.; Oturai, N. B.; Syberg, K.; Hansen, S. F., How environmental regulation can drive innovation: Lessons learned from a systematic review. Environmental Policy and Governance 2022,
  2. Borsatto, J. M. L. S.; Bazani, C. L., Green innovation and environmental regulations: A systematic review of international academic works. Environmental Science and Pollution Research 2021, 1-18
  3. Neves, S. A.; Marques, A. C.; Patrício, M., Determinants of CO2 emissions in European Union countries: does environmental regulation reduce environmental pollution? Economic Analysis and Policy 2020, 68, 114-125
  4. Altenburg, T.; Engelmeier, T., Boosting solar investment with limited subsidies: Rent management and policy learning in India. Energy Policy 2013, 59, 866-874
  5. Harrison, A.; Martin, L. A.; Nataraj, S., Green industrial policy in emerging markets. Annual Review of Resource Economics 2017, 9, 253-274
  6. Wong, C. W.; Wong, C. Y.; Boon-itt, S., Environmental management systems, practices and outcomes: Differences in resource allocation between small and large firms. International Journal of Production Economics 2020, 228, 107734
  7. Wong, C. Y.; Boon-itt, S.; Wong, C. W., The contingency effects of internal and external collaboration on the performance effects of green practices. Resources, Conservation and Recycling 2021, 167, 105383
  8. Bougheas, S.; Mateut, S.; Mizen, P., Corporate trade credit and inventories: New evidence of a trade-off from accounts payable and receivable. Journal of Banking & Finance 2009, 33, (2), 300-307
  9. Jory, S. R.; Khieu, H. D.; Ngo, T. N.; Phan, H. V., The influence of economic policy uncertainty on corporate trade credit and firm value. Journal of Corporate Finance 2020, 64, 101671

We have revised our manuscript according to your comments and suggestions.